# OBSERVABLE PROPAGATION: UNCOVERING FEATURE VECTORS IN TRANSFORMERS

## ABSTRACT

A key goal of current mechanistic interpretability research in NLP is to find *linear features* (also called "feature vectors") for transformers: directions in activation space corresponding to concepts that are used by a given model in its computation. Present state-of-the-art methods for finding linear features require large amounts of labelled data – both laborious to acquire and computationally expensive to utilize. In this work, we introduce a novel method, called "observable propagation" (in short: OBPROP), for finding linear features used by transformer language models in computing a given task – *using almost no data*. Our paradigm centers on the concept of "observables", linear functionals corresponding to given tasks. We then introduce a mathematical theory for the analysis of feature vectors: we provide theoretical motivation for why LayerNorm nonlinearities do not affect the direction of feature vectors; we also introduce a similarity metric between feature vectors called the *coupling coefficient* which estimates the degree to which one feature's output correlates with another's. We use OBPROP to perform extensive qualitative investigations into several tasks, including gendered occupational bias, political party prediction, and programming language detection. Our results suggest that OBPROP surpasses traditional approaches for finding feature vectors in the low-data regime, and that OBPROP can be used to better understand the mechanisms responsible for bias in large language models.

## 1 INTRODUCTION

When a large language model predicts that the next token in a sentence is far more likely to be "him" than "her", what is causing it to make this decision? The field of mechanistic interpretability aims to answer such questions by investigating how to decompose the computation carried out by a model into human-understandable pieces. This helps us predict their behavior, identify and correct discrepancies, align them with our goals, and assess their trustworthiness, especially in high-risk scenarios. The primary goal is to improve output prediction on real-world data distributions, identify and understand discrepancies between intended and actual behavior, align the model with our objectives, and assess trustworthiness in high-risk applications (Olah et al., 2018).

One important notion in mechanistic interpretability is that of "features". A feature can be thought of as a simple function of the activations at a particular layer of the model, the value of which is important for the model's computation at that layer. For instance, in the textual domain, features used by a language model at some layer might reflect whether a token is an adverb, whether the language of the token is French, or other such characteristics. Possibly the most sought-after type of feature is a "linear feature", or "feature vector": a fixed vector in embedding space that the model utilizes by determining how much the input embedding points in the direction of that vector. Linear features are in some sense the holy grails of features: they are both easy for humans to interpret and amenable to mathematical analysis (Olah, 2022).

**Contributions**     Our primary contribution is a method, which we call "observable propagation" (OBPROP in short), for both finding feature vectors in large language models corresponding to given tasks, and analyzing these features in order to understand how they affect other tasks. Unlike non-feature-based interpretability methods such as saliency methods (Simonyan et al., 2013; Jacovi et al., 2021; Wallace et al., 2019) or circuit discovery methods (Conmy et al., 2023; Wang et al., 2022), observable propagation reveals the specific information from the **model's internal activations** that

are responsible for its output, rather than merely tokens or model components that are relevant. And unlike methods for finding feature vectors such as probing (Gurnee et al., 2023; Li et al., 2023; Elazar et al., 2021) or sparse autoencoders Cunningham et al. (2023), observable propagation can find these feature vectors without having to store large datasets of embeddings, perform many expensive forward passes, or utilize vast quantities of labeled data. In addition, we present the following contributions:

• We develop a detailed theoretical analysis of feature vectors. In Theorem 1, we provide theoretical motivation explaining why LayerNorm sublayers do not affect the direction of feature vectors; making progress towards answering the question of the extent to which LayerNorms are used in computation in transformers, which has been raised in mechanistic interpretability (Winsor, 2022). In Theorem 2, we introduce and motivate a measurement of feature vector similarity called the "coupling coefficient", which can be used to determine the extent to which the model's output on one task is coupled with the model's output on another task.

• In order to determine the effectiveness of OBPROP in understanding the causes of bias in large language models, we investigate gendered pronoun prediction (§4.1) and occupational gender bias (§4.2). By using observable propagation, we show that *the model uses the same features to predict gendered pronouns given a name, as it does to predict an occupation given a name*; this is supported by further experiments on both artificial and natural datasets.

• We perform a quantitative comparison between OBPROP and probing methods for finding feature vectors on diverse tasks (subject pronoun prediction, programming language detection, political party prediction). We find that OBPROP is able to achieve superior performance to these traditional data-heavy approaches in low-data regimes (§4.3).

## 1.1 BACKGROUND AND RELATED WORK

In interpretability for NLP applications, there are a number of *saliency-based* methods that attempt to determine which tokens in the input are relevant to the model's prediction (Simonyan et al., 2013; Jacovi et al., 2021; Wallace et al., 2019). Additionally, recent *circuit-based* mechanistic interpretability research has involved determining which components of a model are most relevant to the model's computation on a given task (Conmy et al., 2023; Wang et al., 2022). Our work goes beyond these two approaches by considering not just relevant tokens, and not just relevant model components, but *relevant feature vectors*, which can be analyzed and compared to understand all the intermediate information used by models in their computation. A separate line of research aims to find feature vectors by performing supervised training of probes to find directions in embedding space that correspond to labels (Gurnee et al., 2023; Li et al., 2023; Elazar et al., 2021; Cunningham et al., 2023), or use unsupervised autoencoders on model embeddings to find feature vectors Cunningham et al. (2023). OBPROP does not rely on any training. A number of recent studies in interpretability involve finding feature vectors by decomposing transformer weight matrices into a set of basis vectors and projecting these vectors into token space (Dar et al., 2023; Millidge & Black, 2023). OBPROP goes beyond this by taking into account nonlinearities, by finding precise feature vectors for tasks and by formulating the concept of "observables", which is more general than the tasks considered in these prior works.

## 2 OBSERVABLE PROPAGATION: FROM TASKS TO FEATURE VECTORS

In this section, we present our method, which we call "observable propagation" (OBPROP), for finding feature vectors directly corresponding to a given task. We begin by introducing the concept of "observables", which is central to our paradigm. We then explain observable propagation for simple cases, and then build up to a general understanding.

## 2.1 OUR PARADIGM: OBSERVABLES

Often, in mechanistic interpretability, we care about interpreting the model's computation on a specific task. In particular, the model's behavior on a task can frequently be expressed as the difference between the logits of two tokens. For instance, Mathwin et al. (2023) attempt to interpret the model's understanding of gendered pronouns, and as such, measure the difference between the logits for the

tokens `" she"` and `" he"`. This has been identified as a general pattern of taking "logit differences" that appears in mechanistic interpretability work (Nanda, 2022).

The first insight that we introduce is that each of these logit differences corresponds to a *linear functional on the logits*. That is, if the logits are represented by the vector $y$, then each logit difference can be represented by $n^T y$ for some vector $n$. For instance, if $e_{\text{token}}$ is the one-hot vector with a one in the position corresponding to the token *token*, then the logit difference between `" she"` and `" he"` corresponds to the linear functional $n = (e_{\text{" she"}} - e_{\text{" he"}})$.

We thus define an **observable** to be a linear functional on the logits of a language model. In doing so, we no longer consider logit differences as merely a part of the process of performing an interpretability experiment; rather, we consider the broader class of linear functionals as being objects amenable to study in their own right. As we will see, concretizing observables like this will enable us to find sets of feature vectors corresponding to different observables.

## 2.2 Observable propagation for attention sublayers

First, let us consider a linear model $f(x) = Wx$. Given an observable $n$, we can compute the measurement associated with $n$ as $n^T f(x)$, which is just $n^T Wx$. But now, notice that $n^T Wx = (W^T n)^T x$. In other words, $W^T n$ is a feature vector in the domain, such that the dot product of the input $x$ with the feature vector $W^T n$ directly gives the output measurement $n^T f(x)$.

Next, let us consider how to extend this idea to address attention sublayers in transformers. Attention combines information across tokens. They can be decomposed into two parts: the part that determines from which tokens information is taken (query-key interaction), and the part that determines what information is taken from each token to form the output (output-value). Elhage et al. (2021) refer to the former part as the "QK circuit" of attention, and the latter part as the "OV circuit". Following their formulation, each attention layer can be written as

$$x_j^{l+1} = x_j^l + \sum_{h=1}^{H} \sum_{i=1}^{S} \text{score}_{l,h}(x_i^l, x_j^l) W_{l,h}^{OV} x_i^l$$

where $x_j^l$ is the residual stream for token $j \in \{1, ..., S\}$ at layer $l$, $\text{score}_{l,h}(x_i^l, x_j^l)$ is the attention score at layer $l$ associated with attention head $h \in \{1, ..., H\}$ for tokens $x_i^l$ and $x_j^l$, and $W_{l,h}^{OV}$ is the combined output-value weight matrix for attention head $h$ at layer $l$.

In each term in this sum, the $\text{score}_h(x_i^l, x_j^l)$ factor corresponds to the QK circuit, and the $W_{l,h}^{OV} x_i^l$ factor corresponds to the OV circuit. Note that the primary nonlinearity in attention layers comes from the computation of the attention scores, and their multiplication with the $W_{l,h}^{OV} x_i^l$ terms. As such, as Elhage et al. (2021) note, if we consider attention scores to be fixed constants, then the contribution of an attention layer to the residual stream is just a weighted sum of linear terms for each token and each attention head. This means that if we restrict our analysis to the OV circuit, then we can find feature vectors using the method described for linear models. While this restricts the scope of computation, analyzing OV circuits in isolation is still very valuable: doing so tells us *what sort of information, at each stage of the model's computation, corresponds to our observable.* From this point of view, if we have an attention head $h$ at layer $l$, then the direct effect of that attention head on the output logits of the model is proportional to $W_U W_{l,h}^{OV} x_i^l$ for token $i$, where $W_U$ is the model unembedding matrix (which projects the model's final activations into logits space). We thus have that the feature vector corresponding to the OV circuit for this attention head is given by $(W_U W_{l,h}^{OV})^T n$.

This feature vector corresponds to the direct contribution that the attention head has to the output. But an earlier-layer attention head's output can then be used as the input to a later-layer attention head. For attention heads $h, h'$ in layers $l, l'$ respectively with $l < l'$, the computational path starting at token $i$ in layer $l$ is then passed as the input to attention head $h$; the output of this head for that token is then used as the input to head $h'$ in layer $l'$. Then by the same reasoning, the feature vector for this path is: $(W_U W_{l',h'}^{OV} W_{l,h}^{OV})^T n$. Note that this process can be repeated *ad infinitum*.

## 2.3 General form: addressing MLPs and LayerNorms

Along with attention sublayers, transformers also contain nonlinear MLP sublayers and LayerNorm nonlinearities before each sublayer. One main challenge in interpretability for large models has

---

**Algorithm 1** Observable propagation

---

Let $W_U$ be the model unemebdding matrix.
**if** there exists a LayerNorm operation $x \mapsto f(x)$ before the unembedding operation **then**
 $\quad y \leftarrow \nabla \left( n^T W_U f(x) \right) |_{x=x_0}$ for some suitable value of $x_0$
**else**
 $\quad y \leftarrow (W_U)^T n$
**for** $k \in \{|\mathcal{P}|, \dots, 1\}$, starting at the end **do**
 $\quad$ **if** $l_k$ is an attention head **then**
 $\quad\quad$ Let $W_k$ be the OV matrix for $l_k$.
 $\quad\quad y \leftarrow W_k^T y$.
 $\quad$ **if** $l_k$ is an a nonlinearity that maps $x \mapsto f(x)$ **then**
 $\quad\quad y \leftarrow \nabla \left( y^T W_U f(x) \right) |_{x=x_0}$ for some suitable value of $x_0$
**Output:** $y$

---

been the difficulty in understanding the MLP sublayers, due to the polysemantic nature of their neurons (Olah et al., 2020; Elhage et al., 2022). One prior approach to address this is modifying model architecture to increase the interpretability of MLP neurons (Elhage et al., 2022). Instead of architecture modification, we address these nonlinearities by approximating them as linear functions using their first-order Taylor approximations. This approach is reminiscent of that presented by Nanda et al. (2023), who use linearizations of language models to speed up the process of activation patching (Wang et al., 2022); we go beyond this by recognizing that the gradients used in these linearizations act as feature vectors that can be independently studied and interpreted, rather than merely making activation patching more efficient. Taking this into account, the general form of observable propagation, including first-order approximations of nonlinearities, can be implemented as follows. Consider a computational path $\mathcal{P}$ in the model through sublayers $l_1 < l_2 < \cdots < l_k$. Then for a given observable $n$, the feature vector corresponding to sublayer $l$ in $\mathcal{P}$ can be computed according to Algorithm 1. Note that before every sublayer, there is a nonlinear LayerNorm operation. For greatest accuracy, one can find the feature vector corresponding to this LayerNorm by taking its gradient as described above. But as shown in Theorem 1, if one only cares about the directions of the feature vectors and not their magnitudes, then the LayerNorms can be ignored entirely.

## 2.4 THE EFFECT OF LAYERNORMS ON FEATURE VECTORS

LayerNorms are ubiquitous in Transformers, appearing before every MLP and attention sublayer, and before the final unembedding matrix. Therefore, it is worth investigating how they affect feature vectors; if LayerNorms are highly nonlinear, this would cause trouble for OBPROP.

The core LayerNorm function can be defined as $\text{LayerNorm}(x) = \frac{Px}{\|Px\|}$ where $P$ is the orthogonal projection onto the hyperplane orthogonal to $\vec{1}$, the vector of all ones (Brody et al., 2023). Nanda et al. (2023) provide intuition for why we should expect that in high-dimensional spaces, LayerNorm is approximately linear. But it can be shown that the gradient of $\text{LayerNorm}(x)$ is inversely proportional to $\|Px\|$ (see Appendix C.1), so we cannot consider LayerNorm gradients to be constant for inputs of different norms. However, empirically, we found that *LayerNorms had almost no impact on the direction of feature vectors* (see Appendix I). The following statement, which we prove in Appendix H, provides theoretical underpinning for this behavior:

**Theorem 1.** *Define $f(x; n) = n \cdot \text{LayerNorm}(x)$. Define*

$$\theta(x; n) = \arccos \left( \frac{n \cdot \nabla_x f(x; n)}{\|n\| \|\nabla_x f(x; n)\|} \right)$$

*– that is, $\theta(x; n)$ is the angle between $n$ and $\nabla_x f(x; n)$. Then if $n \sim \mathcal{N}(0, I)$ in $\mathbb{R}^d$, and $d \geq 8$ then*

$$\mathbb{E}[\theta(x; n)] < 2 \arccos \left( \sqrt{1 - 1/(d-1)} \right)$$

Note that after every LayerNorm as defined above, the output is multiplied by a fixed scalar constant equal to $\sqrt{d}$ (where $d$ is the embedding diension), multiplied by a learned diagonal matrix, and added

to a learned vector. Thus, the actual operation implemented is $\sqrt{d}W \text{ LayerNorm}(x) + b$, where $W$ is the learned matrix and $b$ is the learned vector. Now, $b$ does not affect the gradient. Additionally, empirically, most of the entries in $W$ tend to be very close to one another (see Appendix E.2), which suggests that we can approximate $W$ as a scalar, meaning that $W$ primarily scales the gradient, rather than changing its direction. Therefore, if we want to analyze the directions of feature vectors rather than their magnitudes, then *we can do so without worrying about LayerNorms*.

## 3 DATA-FREE ANALYSIS OF FEATURE VECTORS

Once we have used this to obtain a given set of feature vectors, we can then perform some preliminary analyses on them, using solely the vectors themselves. This can give us insights into the behavior of the model without having to run forward passes of the model on data.

**Feature vector norms** One technique that can be used to assess the relative importance of model components is investigating the norms of the feature vectors associated with those components. To see why, recall that if $y$ is a feature vector associated with observable $n$ for a model component that implements function $f$, then for an input $x$, we have $n \cdot f(x) = y \cdot x$. Now, if we have no prior knowledge regarding the distribution of inputs to this model component, we can expect $y \cdot x$ to be proportional to $\|y\|$. Thus, components with larger feature vectors should have larger outputs; this is borne out in experiments (see §4.1) Note that when calculating the norm of a feature vector for a computation path starting with a LayerNorm, one must multiply the norm by an estimated norm of the LayerNorm's input (see Appendix E.1 for explanation).

**Coupling coefficients** An important question that we might want to ask about observables is the following: to what extent should we expect inputs that yield high outputs under observable $n_1$ to also yield high outputs for another observable $n_2$? If the outputs under $n_1$ and $n_2$ are highly correlated, then this suggests that the model uses the same underlying features for both observables. Having motivated this problem, let us translate it into the language of feature vectors. If $n_1$ and $n_2$ are observables with feature vectors $y_1$ and $y_2$ for a function $f$, then for inputs $x$, we have $n_1 \cdot f(x) = y_1 \cdot x$ and $n_2 \cdot f(x) = y_2 \cdot x$. Now, if we constrain our input $x$ to have norm $c$, and constrain $x \cdot y_1 = k$, then what is the expected value of $x \cdot y_2$? And what are the maximum/minimum values of $x \cdot y_2$? We present the following theorem to answer both questions:

**Theorem 2.** *Let $y_1, y_2 \in \mathbb{R}^d$. Let $x$ be uniformly distributed on the hypersphere defined by the constraints $\|x\| = s$ and $x \cdot y_1 = k$. Then we have*

$$\mathbb{E}[x \cdot y_2] = k\frac{y_1 \cdot y_2}{\|y_1\|^2}$$

*and the maximum and minimum values of $x \cdot y_2$ are given by*

$$\frac{\|y_2\|}{\|y_1\|}\left(k\cos(\theta) \pm \sin(\theta)\sqrt{s^2\|y_1\|^2 - k^2}\right)$$

*where $\theta$ is the angle between $y_1$ and $y_2$.*

We denote the value $\frac{y_1 \cdot y_2}{\|y_1\|^2}$ by $C(y_1, y_2)$, and call it the "coupling coefficient from $y_1$ to $y_2$". Intuitively, $C(y_1, y_2)$ measures the expected dot product between a vector and $y_2$, assuming that that vector has a dot product of 1 with $y_1$. Additionally, note that Theorem 2 also implies that the coupling coefficient becomes a more accurate estimator as the cosine similarity between $y_1$ and $y_2$ increases. This is borne out experimentally; see §4.1.

## 4 EXPERIMENTS

Armed with our "observable propagation" toolkit for obtaining and analyzing feature vectors, we now turn our attention to the problem of gender bias in LLMs in order to determine the extent to which these tools can be used to diagnose the causes of unwanted behavior.

### 4.1 GENDERED PRONOUNS PREDICTION

We first consider the related question of understanding how a large language model predicts gendered pronouns. Specifically, given a sentence prefix including a traditionally gendered name (for

| Observable | Heads with greatest feature norms | | | | Feature vector norms | | | |
|---|---|---|---|---|---|---|---|---|
| $n_{\text{subj}}$ | 18::11 | **17::14** | **13::11** | **15::13** | 237.3 | 236.2 | 186.4 | 145.4 |
| $n_{\text{obj}}$ | **17::14** | 18::11 | **13::11** | **15::13** | 159.2 | 157.0 | 145.0 | 112.3 |
| Observable | Heads with greatest attributions | | | | Path patching attributions | | | |
| $n_{\text{subj}}$ | **17::14** | **13::11** | **15::13** | 13::3 | 5.004 | 3.050 | 1.199 | 0.584 |
| $n_{\text{obj}}$ | **17::14** | **13::11** | **15::13** | 22::2 | 2.949 | 1.885 | 1.863 | 0.365 |

Table 1: The four attention heads with the greatest feature norms and path patching attributions (corrupted-clean logit differences) for both the $n_{\text{subj}}$ and $n_{\text{obj}}$ observables. $n_{\text{subj}}$ is the observable measuring the difference between the logits for " she" and " he"; $n_{\text{obj}}$ is the observable measuring the difference between the logits for " her" and " him". "$l::k$" denotes the attention head with index $k$ at layer with index $l$. "Feature vector norms" refers to the norm of the feature vector associated with the attention head; "Path patching attributions" refers to the difference between the model's output for the given observable when the given attention head's activations was patched, and the model's output for that given observable when the attention head was not patched.

example, "Mike" is often associated with males and "Jane" is often associated with females), how does the model predict what kind of pronoun should come after the sentence prefix? We will later see that understanding the mechanisms driving the model's behavior on this benign task will yield insights for understanding gender-biased behavior of the model. Additionally, this investigation also provides an opportunity to test the ability of OBPROP to accurately predict model behavior.

The gendered pronoun prediction problem was previously considered by Mathwin et al. (2023), where the authors used the "Automated Circuit Discovery" tool presented by Conmy et al. (2023) to investigate the flow of information between different components of GPT-2-small (Radford et al., 2019) in predicting subject pronouns (i.e. "he", "she", etc). We extend the problem setting in various ways. We investigate both the subject pronoun case (in which the model is to predict the token "she" versus "he") and the object pronoun case (in which the model is to predict "her" versus "him"). Additionally, we seek to understand the underlying features responsible for this task, rather than just the model components involved, so that we can compare these features with the features that the model uses in producing gender-biased output.

**Problem setting**   We consider two observables, corresponding to the subject pronoun prediction task and the object pronoun prediction task. The observable for the subject pronoun task, $n_{\text{subj}}$, is given by $e_{\text{" she"}} - e_{\text{" he"}}$, where $e_{\text{token}}$ is the one-hot vector with a one in the position corresponding to the token *token*. This corresponds to the logit difference between the tokens " she" and " he", and indicates how likely the model predicts the next token to be " she" versus " he". Similarly, the observable for the object pronoun task, $n_{\text{obj}}$, is given by $e_{\text{" her"}} - e_{\text{" him"}}$.

We investigate the model GPT-Neo-1.3B (Black et al., 2021), which has approximately 1.3B parameters, 24 layers, 16 attention heads, an embedding dimension of 2048, and an MLP hidden dimension of 8192. Note that OBPROP is able to work with models that are significantly larger than those previously explored, such as GPT-2-small (117M parameters) (Radford et al., 2019), which has been the focus of recent interpretability work by Wang et al. (2022), inter alia.

Additionally, a note on notation. The attention head with index $h$ at layer $l$ will be presented as "$l::h$". For instance, 17::14 refers to attention head 14 at layer 17. Furthermore, the MLP at layer $L$ will be presented as "mlp$L$". For instance, mlp1 refers to the MLP at layer 1.

**Feature vector norms for single attention heads**   We begin by analyzing the norms for the feature vectors corresponding to $n_{\text{subj}}$ and $n_{\text{obj}}$ for each attention head in the model. We then used path patching (Goldowsky-Dill et al., 2023) to measure the mean degree to which each attention head contributes to the model's output on dataset of male/female prompt pairs. If our method is effective, then we would expect to see that the heads with the greatest feature norms are those identified by path patching as most important to model behavior. The results are given in Table 1.

We see that three of the four attention heads with the highest feature norms – that is, 17::14, 15::13, and 13::11 – also have very high attributions for both the subject and object pronoun cases. (Interest-

| Head | Coupling coefficient | Cosine similarity | Best-fit slope | $r^2$ |
|------|---------------------|-------------------|----------------|-------|
| 17::14 | 0.7123 | 0.9882 | 0.7692 | 0.9567 |
| 15::13 | 0.8011 | 0.9816 | 0.8003 | 0.9523 |
| 13::11 | 0.7478 | 0.9352 | 0.7632 | 0.8189 |
| 6::6→... | – | 0.9521 | – | 0.8613 |

Table 2: Coupling coefficients and cosine similarity, compared to the slope of the best-fit line for empirical dot products with feature vectors of $n_{\text{subj}}$ versus $n_{\text{obj}}$. Note that for the 6::6→... feature vectors, we do not investigate coupling coefficients, because these earlier-layer attention heads are involved in many computational paths, so the magnitudes obtained for these feature vectors along one computational path do not reflect the importance along the sum total of computational paths.

ingly, head 18::11 does not have a high attribution in either case despite having a large feature norm; this may be due to effects involving the model's QK circuit.) This indicates that observable propagation was largely successful in being able to predict the most important attention heads, despite only using one forward pass per observable (to estimate LayerNorm gradients).

**Cosine similarities and coupling coefficients**   Next, we investigated the cosine similarities between feature vectors for $n_{\text{subj}}$ and $n_{\text{obj}}$. We found that the four heads with the highest cosine similarities between its $n_{\text{subj}}$ feature vector and its $n_{\text{obj}}$ feature vector are 17::14, 18::11, 15::13, and 13::11, with cosine similarities of 0.9882, 0.9831, 0.9816, 0.9352. The high cosine similarities of these feature vectors indicates that the model uses the same underlying features for both the task of predicting subject pronoun genders and the task of predicting object pronoun genders.

We also looked at the feature vectors for the computational paths 6::6→9::1→13::11 for $n_{\text{subj}}$ and 6::6→13::11 for $n_{\text{obj}}$, because performing path patching on a pair of prompts suggested that these computational paths were relevant. The feature vectors for these paths had a cosine similarity of 0.9521.

We then computed the coupling coefficients between the $n_{\text{subj}}$ and $n_{\text{obj}}$ feature vectors for heads 17::14, 15::13, and 13::11. This is because these heads were present among the heads with the highest cosine similarities, highest feature norms, and highest patching attributions, for both the $n_{\text{subj}}$ and $n_{\text{obj}}$ cases. After this, we tested the extent to which the coupling coefficients accurately predicted the constant of proportionality between the dot products of different feature vectors with their inputs. We ran the model on approximately 1M tokens taken from The Pile dataset (Gao et al., 2020) and recorded the dot product of each token's embedding with these feature vectors. We then computed the least-squares best fit line that predicts the $n_{\text{obj}}$ values given the $n_{\text{subj}}$ values, and compared the slope of the line to the coupling coefficients. The results are given in Table 2. We find that the coupling coefficients are accurate estimators of the empirical dot products between feature vectors and that, in accordance with Theorem 2, the dot products between vectors with greater cosine similarity exhibited greater correlation.

## 4.2   OCCUPATIONAL GENDER BIAS

Now that we have understood some of the features relevant to predicting gendered pronous, we more directly consider the setting of occupational gender bias in language models, a widely-investigated problem (Bolukbasi et al., 2016; Vig et al., 2020). For a prompt like `"My friend [NAME] is an excellent ..."`, an LM which hasn't been aligned using e.g. RLHF (Ouyang et al., 2022) is more likely to predict that the next token is `" programmer"` than `" nurse"` if `[NAME]` is replaced with a male name, and vice-versa for a female name (Brown et al., 2020). We applied observable propagation to the problem in order to go beyond prior work and understand the features responsible for this behavior. In particular, we considered the observable $n_{\text{bias}} = (e_{\text{" nurse"}} + e_{\text{" teacher"}} + e_{\text{" secretary"}}) - (e_{\text{" programmer"}} + e_{\text{" engineer"}} + e_{\text{" doctor"}})$; this observable represents the extent to which the model predicts stereotypically-female occupations instead of stereotypically-male ones.

**The same features are used to predict gendered pronouns and occupations**   We ran path patching on a single pair of prompts in order to determine computational paths relevant to $n_{\text{bias}}$. The results were computational paths beginning with mlp1→6::6→9::1→... and 6::6→9::1→..., which began

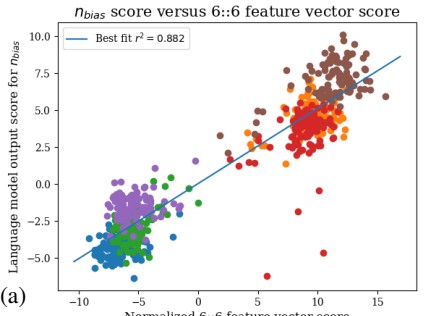
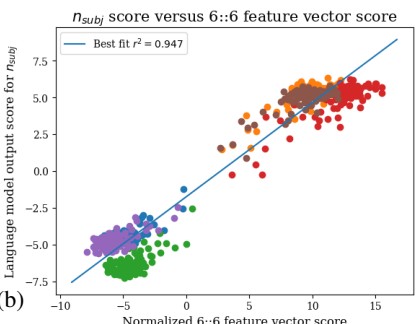

Figure 1: The dot product of model activations with (normalized) feature vectors, compared to the model's output for observables. (a) Dot products with the $n_\text{bias}$ feature vector for 6::6→9::1→..., versus the model's output with respect to $n_\text{bias}$. (b) Dot products with the $n_\text{subj}$ feature vector for 6::6→9::1→13::11, versus the model's output w.r.t $n_\text{subj}$.

on the token in the prompt associated with the gendered name. Even though there were many relevant computational paths beginning with these prefixes, and even though these computational paths passed through multiple later-layer MLPs, the feature vectors for these different paths nevertheless had high cosine similarity with one another.

More surprising is that the feature vector for $n_\text{bias}$ for 6::6→9::1→... had a cosine similarity of 0.966 with the feature vector for $n_\text{subj}$ for 6::6→9::1→13::11. Similarly, the $n_\text{bias}$ feature vector for mlp1→6::6→9::1→... had a cosine similarity of 0.977 with the $n_\text{subj}$ feature vector for mlp1→6::6→9::1→13::11. This indicates that *the model uses the same features to identify both gender pronouns and likely occupations, given a traditionally-gendered name*.

To determine the extent to which these feature vectors reflected model behavior, we ran the model on an artificial dataset of 600 prompts involving gendered names (see Appendix B), recorded the dot product of the model's activations on the name token with the feature vectors, and recorded the model's output with respect to the observables. The results can be found in Figure 1. Note that the correlation coefficient $r^2$ between the dot product with the $n_\text{bias}$ feature vector and the actual model output is 0.88, indicating that the feature vector is a very good predictor of model output.

We then investigated the tokens in a 1M-token subset of The Pile that maximally activated the $n_\text{bias}$ feature vector. These tokens were primarily female names: tokens like `" Rita"`, `" Catherine"`, and `" Mary"`, along with female name suffixes like `"a"` (as in "Phillipa"), `"ine"` (as in "Josephine"), and `"ia"` (as in "Antonia"). Surprisingly, the least-activating tokens were generally male common nouns, such as `" husband"`, `" brother"`, and `" son"` – but also words like `" his"`, and even `" male"`. This evidence even further supports the hypothesis that the model specifically uses gendered features in order to determine which occupations are most likely to be associated with a name. However, it is worth noting that part of the power of OBPROP is that it allows us to test hypotheses such as this without needing to run the model on large datasets and record the tokens with the highest feature vector activations: simply by virtue of the extremely high cosine similarity between the $n_\text{subj}$ feature vector and the $n_\text{bias}$ feature vector, we could infer that the model was using gendered information to predict occupations. As such, looking at the maximally-activating tokens primarily served as a "sanity check", verifying that the feature vectors returned by OBPROP are human-interpretable.

### 4.3 QUANTITATIVE ANALYSIS ACROSS OBSERVABLES

We now evaluate OBPROP's performance across a broader variety of tasks, including subject pronoun prediction, identifying American politicians' party affiliations, and distinguishing between C and Python code. We compare the performance of these feature vectors to the performance of feature vectors obtained by linear/logistic regression, a more traditional method (used by e.g. Kim et al. (2018) and others), but a more data-intensive one. For the former two tasks, we evaluate the correlation between the feature vectors and model outputs on the aforementioned artificial dataset used in the subject pronoun prediction experiments, and on an artificial dataset comprised of 40 Democratic politicians and 40 Republican politicians. For the programming language classification

| Task | OBPROP | Logistic regression (trained) | Regression training set size |
|------|--------|------------------------------|------------------------------|
| Subject pronoun prediction | $r^2 \approx 0.945$ | $r^2 \approx 0.945$ | 60 prompts |
| Political party prediction | $r^2 \approx 0.427$ | $r^2 \approx 0.295$ | 60 prompts (3/4 of dataset) |
| C vs. Python | AUC $\approx 0.9974$ | AUC $\approx 0.9971$ | 50 code snippets |

Table 3: Accuracy of regression-derived feature vectors vs. OBPROP feature vectors.

task, we evaluate the effectiveness of feature vectors in differentiating C and Python code using a natural dataset and the "Area Under the ROC Curve" metric.

The results are given in Table 3. For the subject pronoun prediction task, in order for the feature vector found by linear regression to match the performance of the OBPROP feature vector, 60 prompts' worth of embeddings had to be used for training; similarly, for the C vs. Python classification task, the logistic regression had to be trained on 50 code snippets' worth of embeddings to obtain equal performance. In the political party prediction task, even when training on 3/4 of the dataset, the linear regression feature vector's performance on the test set was well below that of the OBPROP feature vector's performance on the whole dataset. This suggests the ability of OBPROP to match the performance of prior methods for finding feature vectors, and outcompete them especially in the low-data regime.

## 5    CONCLUSION AND DISCUSSION

In this paper, we introduced observable propagation (or OBPROP for short), a novel method for finding feature vectors in transformer models using little to no data. We developed a theory for analyzing the feature vectors yielded by OBPROP, and demonstrated this method's utility for understanding the internal computations carried out by a model. In our case studies, we found that investigating the norms of feature vectors obtained via OBPROP could be used to predict relevant attention heads for a task without actually running the model on any data; that OBPROP can be used to understand when two different tasks utilize the same feature; that coupling coefficients can be used to show the extent to which a high output for one observable implies a high output for another on a general distribution of data; and that the feature vectors returned by OBPROP accurately predict model behavior. We also demonstrated that in data-scarce settings, OBPROP outperforms traditional data-heavy probing approaches for finding feature vectors.

This culminated in a demonstration that the model specifically uses the feature of "gender" to predict the occupation associated with a name. Notably, even though experiments on larger datasets further supported this claim, observable propagation alone was able to provide striking evidence of this using minimal amounts of data. We hope that our approach, being independent of data, can democratize interpretability research and facilitate broader-scale investigations.

Furthermore, the conclusion that the model uses the same mechanisms to predict grammatical gender as it does to predict occupations portends difficulties in attempting to "debias" the model. This means that inexpensive inference-time attempts to remove bias from the model will likely also decrease model performance on desired tasks like correct gendered pronoun prediction (see Appendix F for additional experiments.) This reveals a clear future work direction to invest in more powerful methods, to ensure that models are both unbiased and useful.

Note that although OBPROP demonstrates significant promise in cheaply unlocking the internal computations of language models, it does have limitations. In particular, OBPROP only addresses the OV circuits of Transformers, ignoring computations in QK circuits responsible for mechanisms such as "induction heads" (Elhage et al., 2021). However, even though QK circuits are responsible for moving information around in Transformers, OV circuits are where computation on this information occurs. Thus, whenever we want to understand what sort of information the model uses to predict one token as opposed to another, the answer to this question lies in the model's OV circuits, and OBPROP can provide such answers. Given the power that the current formulation of OBPROP has demonstrated already in our experiments, we are very excited about the potential for this method, and methods building upon it, to yield even greater insights in the near future.

**Ethics statement** In this work, we present observable propagation, our method for finding feature vectors used by large language models in their computation of a given task. We demonstrate in an experiment that observable propagation can be used to pin-point specific features that are responsible for gender bias in large language models, suggesting that observable propagation might prove to be useful in mechanistically understanding how to debias language models. Additionally, the data-efficient nature of observable propagation allows this sort of inquiry into model bias to be democratized, conducted by researchers who might not have access to compute or data required by other methods. However, it is important to note that observable propagation does not necessarily make perfect judgments about model bias or lack thereof; a model might be biased even if observable propagation fails to find specific feature vectors responsible for that bias. As such, it is incumbent upon researchers, practitioners, and organizations working with large language models to continue to perform deeper investigations into model bias issues, and be aware of the way in which it might affect their results.

**Reproducibility statement** A proof of Theorem 1 is given in Appendix H; a proof of Theorem 2 is given in Appendix J. Details on the datasets that we used in our experiments can be found in Appendix B. Further details regarding the experiments in Section 4.3 can be found in Appendix G. Details on how we chose the $x_0$ point used to approximate nonlinearities (as described in §2.3) can be found in Appendix D; for LayerNorm linear approximations, we used the estimation method described in Appendix C.1. We plan to release code soon.

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

## A  FORMAL DEFINITION OF OBSERVABLES

For further clarification, in this section, we provide a more formal definition of an observable.

**Definition A.1.** An **observable** is a linear functional $n : \mathbb{R}^{\texttt{d\_vocab}} \to \mathbb{R}$, where `d_vocab` is the number of tokens in the model's vocabulary. We refer to the action of taking the inner product of the model's output logits with an observable $n$ as *getting the output of the model under the observable $n$*.

Note that because all observables are linear functionals on a finite vector space, they can be written as row vectors. As such, it is often convenient to abuse notation, and associate an observable with its corresponding vector.

Observables often correspond to tasks on which we want to measure model output. The following example demonstrates how observables corresponding to a specific task can be constructed.

**Example A.1.** Consider the task of predicting gendered subject pronouns. We want to measure the difference between the model's predicted logits for the pronoun "she" and the pronoun "he"; this corresponds to how much more likely the model thinks that the next token will be the female subject pronoun over the male subject pronoun. Then if $e_{\texttt{token}}$ is the one-hot vector with size d_vocab and a one in the position corresponding to token, then an observable corresponding to this task is given by $n = e_{"\texttt{she}"} - e_{"\texttt{he}"}$. This is because the output of the model under this observable precisely corresponds to the desired logit difference.

# B DATASETS

In our experiments, we made use of an artificial dataset, along with a natural dataset. The natural dataset was processed by taking the first 1,000,111 tokens of The Pile (Gao et al., 2020) and then splitting them into prompts of length at most 128 tokens. This yielded 7,680 prompts.

To construct the artificial dataset, we wrote three prompt templates for the $n_{\text{subj}}$ observable, three prompt templates for the $n_{\text{bias}}$ observable, and three prompt templates for the $n_{\text{obj}}$ observable. The prompt templates are as follows:

- Prompt templates for $n_{\text{subj}}$ (inspired by Mathwin et al. (2023)):
  1. `"<|endoftext|>So, [NAME] really is a great friend, isn't"`
  2. `"<|endoftext|>Man, [NAME] is so funny, isn't"`
  3. `"<|endoftext|>Really, [NAME] always works so hard, doesn't"`
- Prompt templates for $n_{\text{obj}}$:
  1. `"<|endoftext|>What do I think about [NAME]? Well, to be honest, I love"`
  2. `"<|endoftext|>When it comes to [NAME], I gotta say, I really hate"`
  3. `"<|endoftext|>This is a present for [NAME]. Tomorrow, I'm gonna give it to"`
- Prompt templates for $n_{\text{bias}}$:
  1. `"<|endoftext|>My friend [NAME] is an excellent"`
  2. `"<|endoftext|>Recently, [NAME] has been recognized as a great"`
  3. `"<|endoftext|>His cousin [NAME] works hard at being a great"`

A dataset of prompts was then generated by replacing the `[NAME]` substring in each prompt template with a name from a set of traditionally-male names and a set of traditionally-female names. These names were obtained from the "Gender by Name" dataset from UCI Machine Learning Repository (2020), which provided a list of names, the gender traditionally associated with each name, and a measure of the frequency of each name. The top 100 single-token traditionally-male names and top 100 single-token traditionally-female names from this dataset were collected; this comprised the list of names that we used.

# C MORE ON LAYERNORM GRADIENTS

## C.1 LAYERNORM GRADIENTS ARE INVERSELY PROPORTIONAL TO INPUT NORMS

In §2.4, it was stated that LayerNorm gradients are not constant, but instead, depend on the norm of the input to the LayerNorm. To elaborate, the gradient of $n^T(\sqrt{d}W \text{LayerNorm}(x) + b)$ can be shown to be $\frac{\sqrt{d}W}{\|Px\|} P \left( I - \frac{(Px)(Px)^T}{\|Px\|^2} \right) n$ (see Appendix H). $P$ and $\left( I - \frac{(Px)(Px)^T}{\|Px\|^2} \right)$ are both orthogonal projections that leave $\|n\|$ relatively untouched, so the term that is most responsible for

affecting the norm of the feature vector is the $\frac{\sqrt{d}W}{\|Px\|}$ factor. Now, by Lemma 1 in Appendix H, we have that $\frac{\sqrt{d}W}{\|Px\|} \approx \frac{\sqrt{d}W}{\|x\|}$. Thus, if $\widetilde{\|x\|}$ a good estimate of $\|x\|$ for a given set of input prompts at a given layer, then a good approximation of the gradient of a LayerNorm sublayer is given by $\left(\sqrt{d}W/\widetilde{\|x\|}\right) n$. This approximation can be used to speed up the computation of gradients for LayerNorms.

### C.2 FEATURE VECTOR NORMS WITH LAYERNORMS

In §4.1, we explained that looking at the norms of feature vectors can provide a fast and reasonable guess of which model components will be the most important for a given task. However, there is a caveat that must be taken into account regarding LayerNorms. As shown in Appendix C.1, the gradient of a LayerNorm sublayer is approximately inversely proportional to the norm of the input to the LayerNorm sublayer.

Now, assume that we have a computational path beginning at a LayerNorm, where $\widetilde{\|x\|}$ is an estimate of the norm of the inputs to that LayerNorm. Let $y$ be the feature vector for this computational path. Then we have $y \approx \sqrt{d}W/\widetilde{\|x\|}y'$, where $y'$ is the feature vector for the "tail" of the computational path, that comes after the initial LayerNorm.

Given an input $x$, we have that

$$
\begin{aligned}
y \cdot x &\approx \sqrt{d}W/\widetilde{\|x\|}y' \cdot x \\
&= \sqrt{d}\widetilde{\|x\|} \, \|Wy'\| \, \|x\| \cos\theta \\
&\approx \sqrt{d} \, \|Wy'\| \cos\theta
\end{aligned}
$$

Therefore, the dot product of an input vector with the feature vector $y$ will be approximately proportional to $\sqrt{d} \, \|Wy'\|$ – not $\sqrt{d} \, \|Wy'\| \, /\widetilde{\|x\|}$. As such, if one wants to use feature vector norms to predict which feature vectors will have the highest dot products with their inputs, then *that feature vector must not be multiplied by* $1/\widetilde{\|x\|}$.

A convenient consequence of this is that when analyzing computational paths that do not involve any compositionality (e.g. analyzing a single attention head or a single MLP) – then ignoring LayerNorms entirely still provides an accurate idea of the relative importance of attention heads. This is because the only time that a $(\sqrt{d}W/\widetilde{\|x\|})$ term appears with the factor of $1/\widetilde{\|x\|}$ included is for the final LayerNorm before the logits output. As such, since this factor is not dependent on the layer of the component being analyzed, it can be ignored.

## D DETAILS ON LINEAR APPROXIMATIONS FOR MLPs

Finding feature vectors for MLPs is a relatively straightforward application of the first-order Taylor approximation. However, there is a fear that if one takes the gradient at the wrong point, then the local gradient will not reflect well the larger-scale behavior of the MLP. For example, the output of the MLP with respect to a given observable might be *saturated* at a certain point: the gradient at this point might be very small, and might even point in a direction inconsistent with the MLP's gradient in the unsaturated regime.

To alleviate this, we use the following method. Define $g(x) = n^T \mathrm{MLP}(x)$, where $n$ is a given observable. If this observable $n$ represents the logit difference between two tokens, then we should be able to find an input on which this difference is very negative, along with an input on which this difference is very positive. For example, if $n$ represents the logit difference between the token " her" and the token " him", then an input containing a male name should make this difference very negative, and an input containing a female name should cause this difference to be very positive.

Thus, we have two points $x_-$ and $x_+$ such that $g(x_-) < 0$ and $g(x_+) > 0$. Since MLPs are continuous, there therefore must be some point $x^*$ at which $g(x^*) = 0$: a point that lies on the

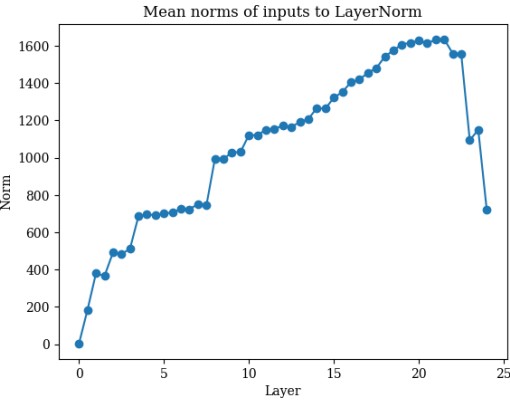

Figure 2: Mean norms of activations before each LayerNorm

decision boundary of the MLP. It stands to reason that the gradient at this decision boundary is more likely to capture the larger-scale behavior of the MLP and is less likely to be saturated, when compared to the gradient at more "extreme" points like $x_-$ and $x_+$.

### D.1 EMPIRICAL MLP APPROXIMATION ACCURACY

In order to evaluate the accuracy of linear approximations of MLP sublayers, we approximated the circuit "MLP16 → MLP19 → unembeddings", and compared the output of this approximation to the output of the actual circuit with respect to $n_{\text{bias}}$ on the artificial dataset described in Appendix B. This circuit was chosen because this is the MLP-containing part of the circuit used to find the $n_{\text{bias}}$ attention-head 6::6 feature vector.

The mean output of this subcircuit with respect to $n_{\text{bias}}$ was approximately 11.887 logits; the root mean squared error of the linear approximation was approximately 0.4139 logits. As such, the root mean squared error was only approximately 3.482% of the mean circuit output. This indicates that, even in this circuit that involves two MLP sublayers, the linear approximation is accurate.

## E EMPIRICAL LAYERNORM GRADIENT INVESTIGATIONS

In this section, we put forth various empirical results relevant for the discussion of LayerNorm gradients in §2.4.

### E.1 LAYERNORM INPUT NORMS PER LAYER

We calculated the average norms of inputs to each LayerNorm sublayer in the model, over the activations obtained from ten of the prompts from the artificial dataset described in §B. The results can be found in Figure 2. The wide variation in the input norms across different layers implies that input norms must be taken into account in any approximation of LayerNorm gradients.

### E.2 LAYERNORM WEIGHT VALUES ARE VERY SIMILAR

In §2.4, we state that the entries in the LayerNorm scaling matrices tend to be very close together, and use this as justification for treating weight matrices as scalars. Specifically, we found that the average variance of scaling matrix entries across all LayerNorms in GPT-Neo-1.3B is 0.007827. To determine the extent to which this variance is large, we calculated the ratio of the variance of each LayerNorm's weight matrix's entries to the mean absolute value of each layer's embeddings' entries. The results can be found in Figure 3. Note that the highest value found was 0.0731 at Layer 0 – meaning that the average entry in Layer 0 embeddings was over 13.67 times larger than the variance between entries in that layer's ln_1 LayerNorm weight. This supports our assertion that LayerNorm scaling matrices can be largely treated as constants.

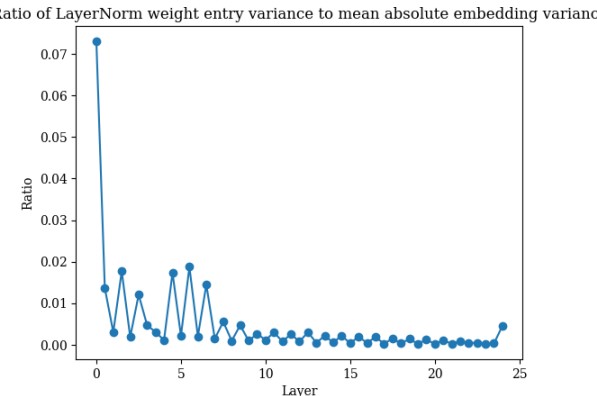

Figure 3: Ratio between LayerNorm weight matrix variances and mean absolute entries of each layer's embeddings

One possible guess as to why this behavior might be occurring is this: much of the computation taking place in the model does not occur with respect to basis directions in activation space. However, the diagonal LayerNorm weight matrices can only act on these very basis directions. Therefore, the weight matrices end up "settling" on the same nearly-constant value in all entries.

## F FURTHER DEBIASING EXPERIMENTS

We ran further experiments on the artificial dataset described in Appendix B, in order to determine the extent to which the feature vectors yielded by observable propagation could be used for debiasing the model's outputs. The idea is similar to that presented by Li et al. (2023): by adding a feature vector to the activations at a given layer for the name token, we can hopefully shift the model's output to be less biased.

Specifically, we used the following methodology. We paired each of the 300 female-name prompts for $n_{\text{bias}}$ with one of the 300 male-name prompts for $n_{\text{bias}}$. For each prompt pair, we ran the model on the female-name prompt and on the male-name prompt, recording the scores with respect to the $n_{\text{bias}}$ observable. We then ran the model on the male-name prompt – but added a multiple of the 6::6 feature vector for $n_{\text{bias}}$ described in §4.2 to the model's activations for the name token before the LayerNorm preceding the layer 6 attention sublayer.

In particular, let $y$ be the unit 6::6 feature vector for $n_{\text{bias}}$, let $x_{\text{female}}$ be the activation vector for the name token at that layer for the female prompt, and let $x_{\text{male}}$ be the activation vector for the name token at that layer for the male prompt. Then we added the vector $y' = ((x_{\text{female}} - x_{\text{male}}) \cdot y) \, y$ to $x_{\text{male}}$. If the model were a linear model whose output was solely determined by the dot product of the input at this layer with the feature vector $y$, then the output of the model in the case where $y'$ is added to the male embeddings would be the same as the output of the model on the female prompt. Therefore, the difference between this "patched" output and the model's output on the female prompt can be viewed as an indicator of the extent to which the feature vector is affected by nonlinearity in the model. We also ran this same experiment, but adding $2y'$ instead of $y'$ to the male embeddings, in order to get a stronger debiasing effect.

The results are given in Table 4. We see that adding $y'$ to the activations for the male prompts is in fact able to cause the model's output to become closer to that of the female prompts – although not as much as it would if the model were linear. But adding $2y'$ to the male prompts' activations is able to bring the model's output to within an average of $1.3180$ logits of the model's output on the female prompts. And when the mean difference between the patched male prompt outputs and the female prompt outputs is calculated without taking the absolute value, this difference becomes even smaller – only $0.1316$ logits on average – which indicates that sometimes, adding $2y'$ to the male prompts' activations even overshoots the model's behavior on the female prompts. As such, we can infer that this feature vector obtained via observable propagation has utility in debiasing the model.

|  | Male prompt | Female prompt | Male patched $+y'$ | Male patched $+2y'$ |
|---|---|---|---|---|
| Mean $n_{\text{bias}}$ score | $-2.947$ | 5.011 | $-0.5489$ | 4.879 |
| Mean absolute difference with female scores | 7.9903 | 0 | 5.6245 | 1.3180 |
| Mean difference from female scores | 7.9583 | 0 | 5.5599 | 0.1316 |

Table 4: The results of the debiasing experiments for the $n_{\text{bias}}$ observable. "Mean absolute difference with female scores" refers to the mean absolute difference between the $n_{\text{bias}}$ score for each male prompt (or male prompt with patched activations) and the score for the corresponding female prompt. "Mean difference from female scores" refers to the mean difference, without taking the absolute value, between the $n_{\text{bias}}$ for the female prompt, and the score for the corresponding (patched) male prompt.

|  | Male prompt | Female prompt | Male patched $+2y'$ |
|---|---|---|---|
| Mean $n_{\text{subj}}$ score | $-5.1393$ | 5.0404 | 4.8794 |
| Mean absolute difference with female scores | 10.180 | 0 | 2.148 |
| Mean difference from female scores | 10.180 | 0 | 0.161 |

Table 5: The results of the debiasing experiments for the $n_{\text{subj}}$ observable, adding $2y'$ to the male prompts' activations.

We then wanted to investigate the extent to which adding this "debiasing vector" would harm the model's performance on the pronoun prediction task. As such, we repeated these experiments on the dataset of prompts for $n_{\text{subj}}$, but adding $2y'$ to the male activations. The results can be found in Table 5. The results show that adding the "debiasing vector" to the male name embeddings also causes the model's ability to correctly predict gendered pronouns to drop dramatically. This suggests that in cases such as this one, where the model uses the same features for undesirable outputs as it does for desirable outputs, inference-time interventions such as that presented by Li et al. (2023) may cause an inevitable decrease in model quality.

## G    EXPERIMENTAL DETAILS FOR SECTION 4.3

### G.1    DATASETS

The dataset used in the subject pronoun prediction task is the same artificial dataset described in Appendix B.

The dataset used in the C vs. Python classification task consists of 730 code snippets, each 128 tokens long, taken from C and Python subsets of the GitHub component of The Pile (Gao et al., 2020).

The dataset used in the American political party prediction task is an artificial dataset consisting of prompts of the form `"[NAME] is a"`, where `[NAME]` is replaced by the name of a politician drawn from a list of 40 Democratic Party politicians and 40 Republican Party politicians. These politicians were chosen according to the list of "the most famous Democrats" and "the most famous Republicans" for Q3 2023 compiled by YouGov, available at https://today.yougov.com/ratings/politics/fame/Democrats/all and https://today.yougov.com/ratings/politics/fame/Democrats/all. The intuition behind this choice of dataset is that the model would be more likely to identify the political affiliation of well-known politicians, because better-known politicians would be more likely to occur in its training data. This is the primary reason that a smaller dataset is being used.

## G.2 TASK DEFINITION

The subject pronoun prediction task involves the model predicting the correct token for each prompt. The target scores are considered to be the difference between the model's logit prediction for the token " she" and the model's logit prediction for the token " he".

The political party prediction task also involves the model predicting the correct token for each prompt. The target scores are considered to be the difference between the model's logit prediction for the token " Democrat" and the model's logit prediction for the token " Republican".

For the C vs. Python classification task, because the data is drawn from a diverse corpus of code, the task is treated as a binary classification task instead of a token prediction task.

## G.3 FEATURE VECTORS

The OBPROP feature vector used for the pronoun prediction task is the feature vector corresponding to the computational path $6::6 \rightarrow 9::1 \rightarrow 13::1$ for the $n_{\text{subj}}$ observable.

The OBPROP feature vector used for the political party prediction task is the feature vector corresponding to attention head 15::8 for the observable defined by $e_{\text{" Democrat"}} - e_{\text{" Republican"}}$.

The OBPROP feature vector used for the C versus Python classification task is the feature vector corresponding to attention head 16::9 for the observable defined by $e_{\text{" ):"}} - e_{\text{" ){"}}$. (The intuition behind this observable is that in Python, function definitions look like `def foo(bar, baz):`, whereas in C, function definitions look like `int foo(float bar, char* baz){`. Notice how the former line ends in the token `"):"` whereas the latter line ends in the token `"){"`.)

The regression feature vectors for each task were trained on model embeddings at the same layer as the OBPROP feature vectors for that task. Thus, for example, the linear regression feature vector for the pronoun prediction task was trained on model embeddings at layer 6.

## G.4 TASK EVALUATION

For the pronoun prediction task, the predicted score was determined as the dot product of the feature vector with the model's embedding at layer 6 for the name token in the prompt.

For the political party prediction task, the predicted score was determined as the dot product of the feature vector with the model's embedding at layer 15 for the last token in the politician's name in each prompt.

For the C versus Python classification task, the predicted score for each code snippet was determined by taking the mean of the model's embeddings at layer 16 for all tokens in the code snippet, and then taking the dot product of the feature vector with those mean embeddings.

## H PROOF OF THEOREM 1

**Theorem 1.** Define $f(x; n) = n \cdot \text{LayerNorm}(x)$. Define

$$\theta(x; n) = \arccos\left(\frac{n \cdot \nabla_x f(x; n)}{\|n\| \, \|\nabla_x f(x; n)\|}\right)$$

– that is, $\theta(x; n)$ is the angle between $n$ and $\nabla_x f(x; n)$. Then if $n \sim \mathcal{N}(0, I)$ in $\mathbb{R}^d$, and $d \geq 8$ then

$$\mathbb{E}\left[\theta(x; n)\right] < 2 \arccos\left(\sqrt{1 - \frac{1}{d-1}}\right)$$

To prove this, we will introduce a lemma:

**Lemma 1.** *Let $y$ be an arbitrary vector. Let $A = I - \frac{vv^T}{\|v\|^2}$ be the orthogonal projection onto the hyperplane normal to $v$. Then the cosine similarity between $y$ and $Ay$ is given by $\sqrt{1 - \cos(\theta)^2}$, where $\cos(\theta)$ is the cosine similarity between $y$ and $v$.*

*Proof.* Assume without loss of generality that $y$ is a unit vector. (Otherwise, we could rescale it without affecting the angle between $y$ and $v$, or the angle between $y$ and $Ay$.)

We have $Ay = y - \frac{y \cdot v}{\|v\|^2} v$. Then,

$$
\begin{aligned}
y \cdot Ay &= y \cdot (y - \frac{y \cdot v}{\|v\|^2} v) \\
&= \|y\|^2 - \frac{(y \cdot v)^2}{\|v\|^2} \\
&= 1 - \frac{(y \cdot v)^2}{\|v\|^2}
\end{aligned}
$$

and

$$
\begin{aligned}
\|Ay\|^2 &= (y - \frac{y \cdot v}{\|v\|^2} v) \cdot (y - \frac{y \cdot v}{\|v\|^2} v) \\
&= y \cdot (y - \frac{y \cdot v}{\|v\|^2} v) - \frac{y \cdot v}{\|v\|^2} v \cdot (y - \frac{y \cdot v}{\|v\|^2} v) \\
&= y \cdot Ay - \frac{y \cdot v}{\|v\|^2} v \cdot (y - \frac{y \cdot v}{\|v\|^2} v) \\
&= y \cdot Ay - \frac{(y \cdot v)^2}{\|v\|^2} + \left\| \frac{y \cdot v}{\|v\|^2} v \right\|^2 \\
&= y \cdot Ay - \frac{(y \cdot v)^2}{\|v\|^2} + \frac{(y \cdot v)^2}{\|v\|^4} \|v\|^2 \\
&= y \cdot Ay - \frac{(y \cdot v)^2}{\|v\|^2} + \frac{(y \cdot v)^2}{\|v\|^2} \\
&= y \cdot Ay
\end{aligned}
$$

Now, the cosine similarity between $y$ and $Ay$ is given by

$$
\begin{aligned}
\frac{y \cdot Ay}{\|y\| \|Ay\|} &= \frac{y \cdot Ay}{\|Ay\|} \\
&= \frac{\|Ay\|^2}{\|Ay\|} \\
&= \|Ay\|
\end{aligned}
$$

At this point, note that $\|Ay\| = \sqrt{y \cdot Ay} = \sqrt{1 - \frac{(y \cdot v)^2}{\|v\|^2}}$. But $\frac{y \cdot v}{\|v\|}$ is just the cosine similarity between $y$ and $v$. Now, if we denote the angle between $y$ and $v$ by $\theta$, we thus have

$$
\|Ay\| = \sqrt{1 - \frac{(y \cdot v)^2}{\|v\|^2}} = \sqrt{1 - \cos(\theta)^2}.
$$

$\square$

Now, we are ready to prove Theorem 1.

*Proof.* First, as noted by Brody et al. (2023), we have that $\text{LayerNorm}(x) = \frac{Px}{\|Px\|}$, where $P = I - \frac{1}{d} \vec{1}\vec{1}^T$ is the orthogonal projection onto the hyperplane normal to $\vec{1}$, the vector of all ones. Thus, we have

$$
f(x; n) = n^T \left( \frac{Px}{\|Px\|} \right)
$$

Using the multivariate chain rule along with the rule that the derivative of $\frac{x}{\|x\|}$ is given by $\frac{I}{\|x\|} - \frac{xx^T}{\|x\|^3}$ (see §2.6.1 of Petersen & Pedersen (2012)), we thus have that

$$
\begin{aligned}
\nabla_x f(x; n) &= \left( n^T \left( \frac{I}{\|Px\|} - \frac{(Px)(Px)^T}{\|Px\|^3} \right) P \right)^T \\
&= \left( \frac{1}{\|Px\|} n^T \left( I - \frac{(Px)(Px)^T}{\|Px\|^2} \right) P \right)^T \\
&= \frac{1}{\|Px\|} P \left( I - \frac{(Px)(Px)^T}{\|Px\|^2} \right) n \qquad \text{because } P \text{ is symmetric}
\end{aligned}
$$

Denote $Q = I - \frac{(Px)(Px)^T}{\|Px\|^2}$. Note that this is an orthogonal projection onto the hyperplane normal to $Px$. We now have that $\nabla_x f(x; n) = \frac{1}{\|Px\|} PQn$. Because we only care about the angle between $n$ and $\nabla_x f(x; n)$, it suffices to look at the angle between $n$ and $PQn$, ignoring the $\frac{1}{\|Px\|}$ term.

Denote the angle between $n$ and $PQn$ as $\theta(x, n)$. (Note that $\theta$ is also a function of $x$ because $Q$ is a function of $x$.) Then if $\theta_Q(x, n)$ is the angle between $n$ and $Qn$, and $\theta_P(x, n)$ is the angle between $Qn$ and $PQn$, then $\theta(x, n) \le \theta_Q(x, n) + \theta_P(x, n)$, so $\mathbb{E}[\theta(x, n)] \le \mathbb{E}[\theta_Q(x, n)] + \mathbb{E}[\theta_P(x, n)]$.

Using Lemma 1, we have that $\theta_Q(x, n) = \arccos\left( \sqrt{1 - \cos(\phi(n, Px))^2} \right)$, where $\phi(n, Px)$ is the angle between $n$ and $Px$. Now, because $n \sim \mathcal{N}(0, I)$, we have $\mathbb{E}[\cos(\phi(n, Px))^2] = 1/d$, using the well-known fact that the expected squared dot product between a uniformly distributed unit vector in $\mathbb{R}^d$ and a given unit vector in $\mathbb{R}^d$ is $1/d$.

At this point, define $g(t) = \arccos\left( \sqrt{1-t} \right)$, $h(t) = g'\left( \frac{1}{d-1} \right) \left( t - \frac{1}{d-1} \right) + g\left( \frac{1}{d-1} \right)$. Then if $\frac{1}{d-1} < c$, where $c$ is the least solution to $g'(c) = \frac{\pi - 2g(c)}{2(1-c)}$, then $h(t) \ge g(t)$. (Note that $g(t)$ is convex on $(0, 0.5]$ and concave on $[0.5, 1)$. Therefore, there are exactly two solutions to $g'(c) = \frac{\pi - 2g(c)}{2(1-c)}$. The lesser of the two solutions is the value at which $g'(c)$ equals the slope of the line between $(c, g(c))$ and $(1, \pi/2)$ – the latter point being the maximum of $g$ – at the same time that $g''(c) \ge 0$.) One can compute $c \approx 0.155241\ldots$, so if $d \ge 8$, then $1/(d-1) < c$ is satisfied, so $h(t) \ge g(t)$. Thus, we have the following inequality:

$$
\begin{aligned}
h(1/(d-1)) &> h(1/d) \\
&= h(\mathbb{E}[\cos(\phi(n, Px))^2]) \\
&= \mathbb{E}[h(\cos(\phi(n, Px))^2)] \text{ due to linearity} \\
&\ge \mathbb{E}[g(\cos(\phi(n, Px))^2)] \text{ because } h(t) \ge g(t) \text{ for all } t \\
&= \mathbb{E}[\theta_Q(x, n)]
\end{aligned}
$$

Now, $h(1/(d-1)) = g(1/(d-1)) = \arccos\left( \sqrt{1 - \frac{1}{d-1}} \right)$. Thus, we have that $\arccos\left( \sqrt{1 - \frac{1}{d-1}} \right) > \mathbb{E}[\theta_Q(x, n)]$.

The next step is to determine an upper bound for $\mathbb{E}[\theta_P(x, n)]$. By Lemma 1, we have that $\theta_P(x, n) = \arccos\left( \sqrt{1 - \cos(\phi(Qn, \vec{1}))^2} \right)$. Now, note that because $n \sim \mathcal{N}(0, I)$, then $Qn$ is distributed according to a unit Gaussian in $\operatorname{Im} Q$, the $(d-1)$-dimensional hyperplane orthogonal to $Px$. Note that because $\vec{1}$ is orthogonal to $Px$ (by the definition of $P$) and $Px$ is orthogonal to $\operatorname{Im} Q$, this means that $\vec{1} \in \operatorname{Im} Q$. Now, let us apply the same fact from earlier: that the expected squared dot product between a uniformly distributed unit vector in $\mathbb{R}^{d-1}$ and a given unit vector in $\mathbb{R}^{d-1}$ is $1/(d-1)$. Thus, we have that $\mathbb{E}[\cos(\phi(Qn, \vec{1}))^2] = 1/(d-1)$.

From this, by the same logic as in the previous case, $\arccos\left( \sqrt{1 - \frac{1}{d-1}} \right) \ge \mathbb{E}[\theta_P(x, n)]$.

| Task | Cosine similarity | Angle (radians) |
|---|---|---|
| Subject pronoun prediction (attention 6::6) | 0.99779 | 0.0664 |
| C vs. Python | 0.99936 | 0.0358 |
| Political party prediction | 0.99900 | 0.0447 |

Table 6: Cosine similarities between the feature vectors used in Section 4.3, computed with and without LayerNorms

Adding this inequality to the inequality for $\mathbb{E}[\theta_Q(x, n)]$, we have

$$2\arccos\left(\sqrt{1 - \frac{1}{d-1}}\right) > \mathbb{E}[\theta_Q(x, n)] + \mathbb{E}[\theta_P(x, n)] \geq \mathbb{E}[\theta(x, n)]$$

. $\qquad\qquad\qquad\qquad\qquad\qquad\qquad\qquad\qquad\qquad\qquad\qquad\qquad\qquad\qquad$ $\square$

## I    EMPIRICAL RESULTS REGARDING THEOREM 1

Note that Theorem 1 assumes that feature vectors $n$ are normally-distributed, which may not necessarily occur in practice (although given that observables and observable-derived feature vectors are only introduced in this work, it is hard to say whether this is false more generally; more research is necessary, including research on the sorts of observables that practitioners wish to analyze in practice). However, the intention of Theorem 1 is to provide motivation that underpins what we found empirically: namely, that feature vectors computer by taking LayerNorm into account have extremely high cosine similarities with feature vectors computed without taking LayerNorm into account.

In particular, for the feature vectors considered in Section 4.3, these cosine similarities and angles are given in Table 6. For reference, note that the upper bound on the angle between these feature vectors according to Theorem 1 is approximately 0.0442 radians. The feature vectors for subject pronoun prediction have a higher angle between them of 0.0664 radians, but this can be attributed to the fact that the circuit for these feature vectors goes through multiple LayerNorms. Additionally, the angle for the political party prediction feature vector is also slightly higher than the bound predicted by the theorem; but it is worth noting that the theorem predicts a bound on the expected angle, rather than a bound on the maximum angle; this also might be due to the scaling matrix $W$ in the LayerNorm (see Appendix E).

## J    PROOF OF THEOREM 2

**Theorem 2.** Let $y_1, y_2 \in \mathbb{R}^d$. Let $x$ be uniformly distributed on the hypersphere defined by the constraints $\|x\| = s$ and $x \cdot y_1 = k$. Then we have

$$\mathbb{E}[x \cdot y_2] = k\frac{y_1 \cdot y_2}{\|y_1\|^2}$$

and the maximum and minimum values of $x \cdot y_2$ are given by

$$\frac{\|y_2\|}{\|y_1\|}\left(k\cos(\theta) \pm \sin(\theta)\sqrt{s^2\|y_1\|^2 - k^2}\right)$$

where $\theta$ is the angle between $y_1$ and $y_2$.

Before proving Theorem 2, we will prove a quick lemma.

**Lemma 2.** *Let $\mathcal{S}$ be a hypersphere with radius $r$ and center $c$. Then for a given vector $y$, the mean squared distance from $y$ to the sphere, $\mathbb{E}_{s \in \mathcal{S}}[\|y - c\|^2]$, is given by $\|y - c\|^2 + r^2$.*

*Proof.* Without loss of generality, assume that $\mathcal{S}$ is centered at the origin (so $\|y - c\|^2 = \|y\|^2$). Induct on the dimension of the $\mathcal{S}$. As our base case, let $\mathcal{S}$ be the 0-sphere consisting of a point in $\mathbb{R}^1$ at $-r$ and a point at $r$. Then $\mathbb{E}_{s \in \mathcal{S}}[|y - s|^2] = \frac{(y-r)^2 + (y-(-r))^2}{2} = y^2 + r^2$.

For our inductive step, assume the inductive hypothesis for spheres of dimension $d - 2$; we will prove the theorem of spheres of dimension $d - 1$ in an ambient space of dimension $d$. Without loss of generality, let $y$ lie on the x-axis, so that we have $y = [y_1 \quad 0 \quad 0 \quad \dots]^T$. Next, divide $\mathcal{S}$ into slices along the x-axis. Denote the slice at position $x = x_0$ as $\mathcal{S}_{x_0}$. Then $\mathcal{S}_{x_0}$ is a $(d - 2)$-sphere centered at $[x_0 \quad 0 \quad 0 \quad \dots]^T$, and has radius $\sqrt{r^2 - x_0^2}$. Now, by the law of total expectation, $\mathbb{E}_{s \in \mathcal{S}}[\|y - s\|^2] = \mathbb{E}_{-r \leq x \leq r}\left[\mathbb{E}_{s' \in \mathcal{S}_x}\left[\|y - s'\|^2\right]\right]$. We then have that

$$\mathbb{E}_{s' \in \mathcal{S}_x}\left[\|y - s'\|^2\right] = \mathbb{E}\left[(y_1 - x)^2 + s_2^2 + s_3^2 + \cdots\right]$$
$$= (y_1 - x)^2 + \mathbb{E}\left[s_2^2 + s_3^2 + \cdots\right]$$

Once again, $\mathcal{S}_x$ is a $(d - 2)$-sphere defined by $s_2^2 + s_3^2 + \cdots = r^2 - x^2$. This means that by the inductive hypothesis, we have $\mathbb{E}\left[s_2^2 + s_3^2 + \cdots\right] = r^2 - x^2$. Thus, we have

$$\mathbb{E}_{s' \in \mathcal{S}_x}\left[\|y - s'\|^2\right] = (y_1 - x)^2 + r^2 - x^2$$
$$\mathbb{E}_{s' \in \mathcal{S}_x}\left[\|y - s'\|^2\right] = (y_1 - x)^2 + r^2 - x^2$$
$$\mathbb{E}_{s \in \mathcal{S}}[\|y - s\|^2] = \mathbb{E}_{-r \leq x \leq r}\left[(y_1 - x)^2 + r^2 - x^2\right]$$
$$= \frac{1}{2r}\int_{-r}^{r}(y_1 - x)^2 + r^2 - x^2 dx$$
$$= r^2 + y_1^2$$

$\square$

We are now ready to begin the main proof.

*Proof.* First, assume that $\|x\| = 1$. Now, the intersection of the $(d - 1)$-sphere defined by $\|x\| = 1$ and the hyperplane $x \cdot y_1 = k$ is a unit hypersphere of dimension $(d - 2)$, oriented in the hyperplane $x \cdot y_1 = k$, and centered at $c_1 y_1$ where $c_1 = k/\|y_1\|^2$. Denote this $(d - 2)$-sphere as $\mathcal{S}$, and denote its radius by $r$.

Next, define $c_2 = \frac{k}{y_2 \cdot y_1}$. Then $c y_2 \cdot y_1 = k$, so $c_2 y_2$ lies in the same hyperplane as $\mathcal{S}$. Additionally, because $c_1 y_1$ is in this hyperplane, and $c_1 y_1$ is also the normal vector for this hyperplane, we have that the vectors $c_1 y_1$, $c_2 y_2$, and $c_1 y_1 - c_2 y_2$ form a right triangle, where $c_2 y_2$ is the hypotenuse and $c_1 y_1 - c_2 y_2$ is the leg opposite of the angle $\theta$ between $y_1$ and $y_2$. As such, we have that $\|c_1 y_1 - c_2 y_2\| = \sin(\theta)\|c_2 y_2\|$.

Furthermore, we have that $c_1 y_1 \cdot c_2 y_2 = \frac{k^2}{\|y_1\|^2}$, that $\|c_1 y_1\| = \frac{|k|}{\|y_1\|^2}$, and that $\|c_2 y_2\| = \frac{|k|}{\|y_1\| |\cos \theta|}$

We will now begin to prove that the maximum and minimum values of $y_2 \cdot x$ are given by $\frac{\|y_2\|}{\|y_1\|}\left(k\cos(\theta) \pm |\sin(\theta)|\sqrt{s^2\|y_1\|^2 - k^2}\right)$.

To start, note that the nearest point on $\mathcal{S}$ to $c_2 y_2$ and the farthest point on $\mathcal{S}$ from $c_2 y_2$ are located at the intersection of $\mathcal{S}$ with the line between $c_2 y_2$ and $c_1 y_1$.

To see this, let $x_+$ be the at the intersection of $\mathcal{S}$ and the line between $c_2 y_2$ and $c_1 y_1$. We will show that $x_+$ is the nearest point on $\mathcal{S}$ to $c_2 y_2$. Let $x'_+ \in \mathcal{S} \neq x_+$. Then we have the following cases:

- Case 1: $c_2 y_2$ is outside of $\mathcal{S}$. Then $\|c_2 y_2 - c_1 y_1\| = \|c_2 y_2 - x_+\| + \|x_+ - c_1 y_1\|$, because $c_2 y_2$, $x_+$, and $c_1 y_1$ are collinear – so $\|c_2 y_2 - c_1 y_1\| = \|c_2 y_2 - x_+\| + r$ (because $x_+ \in \mathcal{S}$). By the triangle inequality, we have $\|c_2 y_2 - c_1 y_1\| \leq \|c_2 y_2 - x'_+\| + \|x'_+ - c_1 y_1\| = \|c_2 y_2 - x'_+\| + r$. But this means that $\|c_2 y_2 - x_+\| \leq \|c_2 y_2 - x'_+\|$.

- Case 2: $c_2 y_2$ is inside of $\mathcal{S}$. Then $\|c_2 y_2 - c_1 y_1\| = \|x_+ - c_1 y_1\| - \|c_2 y_2 - x_+\|$, because $c_2 y_2$, $x_+$, and $c_1 y_1$ are collinear – so $\|c_2 y_2 - c_1 y_1\| = r - \|c_2 y_2 - x_+\|$. By

the triangle inequality, we have $\left\| x'_+ - c_1 y_1 \right\| \leq \left\| c_2 y_2 - x'_+ \right\| + \left\| c_2 y_2 - c_1 y_1 \right\|$, so $\left\| x'_+ - c_1 y_1 \right\| \leq \left\| c_2 y_2 - x'_+ \right\| + r - \left\| c_2 y_2 - x_+ \right\|$. But since $\left\| x'_+ - c_1 y_1 \right\| = r$, this means that $\left\| c_2 y_2 - x_+ \right\| \leq \left\| c_2 y_2 - x'_+ \right\|$.

A similar argument will show that $x_-$, the farthest point on $\mathcal{S}$ from $c_2 y_2$, is also located at the intersection of $\mathcal{S}$ with the line between $c_2 y_2$ and $c_1 y_1$.

Now, let us find the values of $x_+$ and $x_-$. The line between $c_2 y_2$ and $c_1 y_1$ can be parameterized by a scalar $t$ as $c_1 y_1 + t(c_2 y_2 - c_1 y_1)$. Then $x_+$ and $x_-$ are given by $c_1 y_1 + t^*(c_2 y_2 - c_1 y_1)$, where $t^*$ are the solutions to the equation $\left\| c_1 y_1 + t(c_2 y_2 - c_1 y_1) \right\| = 1$.

We have the following:

$$
\begin{aligned}
1 &= \left\| c_1 y_1 + t(c_2 y_2 - c_1 y_1) \right\| \\
&= \left\| c_1 y_1 \right\|^2 + 2t(c_1 y_1 \cdot (c_2 y_2 - c_1 y_1)) + t^2 \left\| c_2 y_2 - c_1 y_1 \right\|^2 \\
&= \left\| c_1 y_1 \right\|^2 + 2t((c_1 y_1 \cdot c_2 y_2) - \left\| c_1 y_1 \right\|^2) + t^2 \left\| c_2 y_2 \right\|^2 \sin^2 \theta \\
&= \frac{k^2}{\left\| y_1 \right\|^2} + 2t \left( \frac{k^2}{\left\| y_1 \right\|^2} - \frac{k^2}{\left\| y_1 \right\|^2} \right) + t^2 \frac{k^2}{\left\| y_1 \right\|^2 \cos^2 \theta} \sin^2 \theta \\
&= \frac{k^2}{\left\| y_1 \right\|^2} (t^2 \tan^2 \theta + 1)
\end{aligned}
$$

Thus, solving for $t$, we have that $t^* = \frac{\pm \sqrt{\left\| y_1 \right\|^2 - k^2}}{|k| \tan \theta}$. Therefore, we have that

$$
\begin{aligned}
x_+, x_- &= c_1 y_1 + t^*(c_2 y_2 - c_1 y_1) \\
&= c_1 y_1 + \left( \frac{k^2}{\left\| y_1 \right\|^2} (t^2 \tan^2 \theta + 1) \right) (c_2 y_2 - c_1 y_1) \\
&= \frac{k y_1}{\left\| y_1 \right\|^2} + \left( \frac{\pm \sqrt{\left\| y_1 \right\|^2 - k^2}}{|k| \tan \theta} \right) \left( \frac{k y_2}{y_1 \cdot y_2} - \frac{k y_1}{\left\| y_1 \right\|^2} \right) \\
&= k \left[ \frac{y_1}{\left\| y_1 \right\|^2} \pm \left( \frac{\sqrt{\left\| y_1 \right\|^2 - k^2}}{|k| \tan \theta} \right) \left( \frac{y_2}{y_1 \cdot y_2} - \frac{y_1}{\left\| y_1 \right\|^2} \right) \right]
\end{aligned}
$$

$$
\begin{aligned}
y_2 \cdot x_+, y_2 \cdot x_- &= y_2 \cdot k \left[ \frac{y_1}{\|y_1\|^2} \pm \left( \frac{\sqrt{\|y_1\|^2 - k^2}}{|k| \tan\theta} \right) \left( \frac{y_2}{y_1 \cdot y_2} - \frac{y_1}{\|y_1\|^2} \right) \right] \\
&= \frac{ky_1 \cdot y_2}{\|y_1\|^2} \pm \left( \cot\theta \sqrt{\|y_1\|^2 - k^2} \right) \left( \frac{y_2 \cdot y_2}{y_1 \cdot y_2} - \frac{y_1 \cdot y_1}{\|y_1\|^2} \right) \\
&= \frac{ky_1 \cdot y_2}{\|y_1\|^2} \pm \left( \cot\theta \sqrt{\|y_1\|^2 - k^2} \right) \left( \frac{\|y_2\|}{\|y_1\| \cos\theta} - \frac{\|y_2\| \cos\theta}{\|y_1\|} \right) \\
&= \left[ \frac{ky_1 \cdot y_2}{\|y_1\|^2} \pm \left( \cot\theta \sqrt{\|y_1\|^2 - k^2} \right) \frac{\|y_2\|}{\|y_1\|} \left( \frac{1}{\cos\theta} - \cos\theta \right) \right] \\
&= \frac{ky_1 \cdot y_2}{\|y_1\|^2} \pm \left( \cot\theta \sqrt{\|y_1\|^2 - k^2} \right) \frac{\|y_2\|}{\|y_1\|} \sin\theta \tan\theta \\
&= \frac{ky_1 \cdot y_2}{\|y_1\|^2} \pm \frac{\|y_2\|}{\|y_1\|} \sin\theta \sqrt{\|y_1\|^2 - k^2} \\
&= \frac{\|y_2\|}{\|y_1\|} \left( k \cos(\theta) \pm \sin(\theta) \sqrt{\|y_1\|^2 - k^2} \right)
\end{aligned}
$$

We will now prove that $\mathbb{E}\left[y_2 \cdot x\right] = \frac{y_1 \cdot y_2}{\|y_1\|^2}$. Before we do, note that we can also use our value of $t^*$ to determine the squared radius of $\mathcal{S}$. We have that the squared radius of $\mathcal{S}$ is given by

$$
\begin{aligned}
r^2 &= \|t^*(c_2 y_2 - c_1 y_1)\|^2 \\
&= (t^*)^2 \|(c_2 y_2 - c_1 y_1)\|^2 \\
&= (t^*)^2 \sin^2\theta \|c_2 y_2\|^2 \\
&= \frac{\sin^2(\theta)k^2 / \left(\|y_1\|^2 \cos^2\theta\right)}{k^2 \tan\theta} \left(\|y_1\|^2 - k^2\right) \\
&= 1 - \frac{k^2}{\|y_1\|^2}
\end{aligned}
$$

We will use this result soon. Now, on to the main event. Begin by noting that $y_2 \cdot x = \|y_2\| \|x\| \cos(y_2, x) = \|y_2\| \cos(y_2, x)$, where $\cos(y_2, x)$ is the cosine of the angle between $y_2$ and $x$. Now, $\cos(y_2, x) = \operatorname{signum}(c_2) \cos(c_2 y_2, x)$. And we have that $\|x - c y_2\|^2 = \|x\|^2 + \|c y_2\|^2 - 2 \|x\| \|c_2 y_2\| \cos(c y_2, x) = 1 + \|c_2 y_2\|^2 - 2 \|c_2 y_2\| \cos(c_2 y_2, x)$. Therefore, we have

$$
\begin{aligned}
\cos(y_2, x) &= \operatorname{signum}(c_2) \cos(c_2 y_2, x) \\
&= \operatorname{signum}(c_2) \frac{\|x - c_2 y_2\|^2 - 1 - \|c_2 y_2\|^2}{-2 \|c_2 y_2\|} \\
&= \operatorname{signum}(c_2) \frac{1 + \|c_2 y_2\|^2 - \|x - c_2 y_2\|^2}{2 \|c_2 y_2\|}
\end{aligned}
$$

$$
\begin{aligned}
y_2 \cdot x &= \|y_2\| \cos(y_2, x) \\
&= \operatorname{signum}(c_2) \|y_2\| \frac{1 + \|c_2 y_2\|^2 - \|x - c_2 y_2\|^2}{2 \|c_2 y_2\|}
\end{aligned}
$$

$$\mathbb{E}\left[y_2 \cdot x\right] = \mathbb{E}\left[\operatorname{signum}(c_2)\left\|y_2\right\| \frac{1 + \left\|c_2 y_2\right\|^2 - \left\|x - c_2 y_2\right\|^2}{2\left\|c_2 y_2\right\|}\right]$$

$$= \operatorname{signum}(c_2)\left\|y_2\right\| \frac{1 + \left\|c_2 y_2\right\|^2 - \mathbb{E}\left[\left\|x - c_2 y_2\right\|^2\right]}{2\left\|c_2 y_2\right\|}$$

$$= \operatorname{signum}(c_2)\left\|y_2\right\| \frac{1 + \left\|c_2 y_2\right\|^2 - \left(1 - \frac{k^2}{\left\|y_1\right\|^2} + \left\|c_1 y_1 - c_2 y_2\right\|^2\right)}{2\left\|c_2 y_2\right\|}$$

This last line uses Lemma 2: $c_1 y_1$ is the center of $\mathcal{S}$, so the expected squared distance between $c_2 y_2$ and a point on $\mathcal{S}$ is given by $1 - \frac{k^2}{\left\|y_1\right\|^2} + \left\|c_1 y_1 - c_2 y_2\right\|^2$, where $1 - \frac{k^2}{\left\|y_1\right\|^2}$ is the squared radius of $\mathcal{S}$ and $\left\|c_1 y_1 - c_2 y_2\right\|^2$ is the squared distance from $c_2 y_2$ to the center. We can use this lemma because $c_2 y_2$ is in the same hyperplane as $\mathcal{S}$, so we can treat this situation as being set in a space of dimension $d - 1$.

Now, continue to simplify:

$$\mathbb{E}\left[y_2 \cdot x\right] = \operatorname{signum}(c_2)\left\|y_2\right\| \frac{1 + \left\|c_2 y_2\right\|^2 - \left(1 - \frac{k^2}{\left\|y_1\right\|^2} + \left\|c_1 y_1 - c_2 y_2\right\|^2\right)}{2\left\|c_2 y_2\right\|}$$

$$= \operatorname{signum}(c_2)\left\|y_2\right\| \frac{\left\|c_2 y_2\right\|^2 + \frac{k^2}{\left\|y_1\right\|^2} - \sin^2\theta\left\|c_2 y_2\right\|^2}{2\left\|c_2 y_2\right\|}$$

$$= \operatorname{signum}(c_2)\left\|y_2\right\| \frac{\left\|c_2 y_2\right\|^2 \cos^2\theta + \frac{k^2}{\left\|y_1\right\|^2}}{2\left\|c_2 y_2\right\|}$$

$$= \operatorname{signum}(c_2)\left\|y_2\right\| \frac{1}{2}\left(\left\|c_2 y_2\right\| \cos^2\theta + \frac{|k|\cos\theta}{\left\|y_1\right\|}\right)$$

$$= \operatorname{signum}(c_2)\left\|y_2\right\| \frac{1}{2}\left(\frac{|k||\cos\theta|}{\left\|y_1\right\|} + \frac{|k||\cos\theta|}{\left\|y_1\right\|}\right)$$

$$= \operatorname{signum}(c_2)|k| \frac{\left\|y_2\right\|}{\left\|y_1\right\|}|\cos\theta|$$

$$= k \frac{\left\|y_2\right\|}{\left\|y_1\right\|}\cos\theta$$

$$= k \frac{y_1 \cdot y_2}{\left\|y_1\right\|^2}$$

The last thing to do is to note that the above formulas are only valid when $\|x\| = 1$. But if $\|x\| = s$, this is equivalent to the case when $\|x\| = 1$ if we scale $y_1$ and $y_2$ by $s$. Scaling those two vectors by $s$ gives us the final formulas in Theorem 2. □

## K  TOP ACTIVATING TOKENS ON $1\text{M}$ TOKENS FROM THE PILE FOR $n_{\text{BIAS}}$ AND $n_{\text{SUBJ}}$ 6::6 FEATURE VECTORS

In §4.2, in order to confirm that the feature vectors that we found for attention head 6::6 corresponded to notions of gender, we looked at the tokens from a dataset of 1M tokens from The Pile (see Appendix B) that maximally and minimally activated these feature vectors.

### K.1  $n_{\text{BIAS}}$ FEATURE VECTOR

The thirty highest-activating tokens, along with the prompts from which they came, and their scores, are given below:

1. Highest-activating token #1:

- Excerpt from prompt: `"son Tower** and in front of it a beautiful statue of St Edmund by Dame Elisabeth Frink (1976). The rest of the abbey spreads eastward like a r"`
- Token: `"abeth"`
- Score: 18.372

2. Highest-activating token #2:
    - Excerpt from prompt: `" a gorgeous hammerbeam roof and a striking sculpture of the crucified Christ by Dame Elisabeth Frink in the north transept.\n\nThe impressive entrance porch has a"`
    - Token: `"abeth"`
    - Score: 17.388

3. Highest-activating token #3:
    - Excerpt from prompt: `" the elaborate Portuguese silver service or the impressive Egyptian service, a divorce present from Napoleon to Josephine"`
    - Token: `"ine"`
    - Score: 16.815

4. Highest-activating token #4:
    - Excerpt from prompt: `" rocky beach of **Priest's Cove**, while nearby are the ruins of **St Helen's Oratory**, supposedly one of the first Christian chapels built in West Cornwall"`
    - Token: `" Helen"`
    - Score: 16.309

5. Highest-activating token #5:
    - Excerpt from prompt: `", and opened in 1892, this brainchild of his Parisian actress wife, Josephine, was built by French architect Jules Pellechet to display a collection the Bow"`
    - Token: `"ine"`
    - Score: 16.267

6. Highest-activating token #6:
    - Excerpt from prompt: `" the film ⎵Bridget Jones's Diary;⎵ a local house was used as Bridget's parents' home.\n\n1Sights\n\nBroadway TowerTOWER"`
    - Token: `"idget"`
    - Score: 16.171

7. Highest-activating token #7:
    - Excerpt from prompt: `") by his side and a loyal band of followers in support. Arthur went on to slay Rita Gawr, a giant who butchered"`
    - Token: `" Rita"`
    - Score: 16.079

8. Highest-activating token #8:
    - Excerpt from prompt: `" for the fact that Sir Robert Walpole's grandson sold the estate's splendid art collection to Catherine the Great of Russia to stave off debts { those paintings formed the foundation of the"`
    - Token: `" Catherine"`

- Score: 16.039

9. Highest-activating token #9:

    - Excerpt from prompt: `" Highlights include the magnificent gold coach of 1762 and the 1910 Glass Coach (Prince William and Catherine Middleton actually used the 1902 State Landau for their wedding in 2011).\n\n"`
    - Token: `" Catherine"`
    - Score: 15.967

10. Highest-activating token #10:

    - Excerpt from prompt: `" by Canaletto, El Greco and Goya as well as 55 paintings by Josephine herself.  Among the 15,000 other objets d'art are incredible dresses from"`
    - Token: `"ine"`
    - Score: 15.906

11. Highest-activating token #11:

    - Excerpt from prompt: `" looks like something from a children's storybook (a fact not unnoticed by the author Antonia Barber, who set her much-loved fairy-tale _The Mousehole Cat"`
    - Token: `"ia"`
    - Score: 15.582

12. Highest-activating token #12:

    - Excerpt from prompt: `".  Precious little now remains save for a few nave walls, the ruined **St Mary's chapel**, and the crossing arches, which may"`
    - Token: `" Mary"`
    - Score: 15.443

13. Highest-activating token #13:

    - Excerpt from prompt: `".\n\nTrain\n\nThe northern terminus of the Welsh Highland Railway is on St Helen's Rd.  Trains run to Porthmadog (£35 return, 2½"`
    - Token: `" Helen"`
    - Score: 15.374

14. Highest-activating token #14:

    - Excerpt from prompt: `"2\n\n### KING RICHARD III\n\nIt's an amazing story.  Philippa Langley, a member of the Richard III Society, spent four-and-a"`
    - Token: `"a"`
    - Score: 15.358

15. Highest-activating token #15:

    - Excerpt from prompt: `" pit (which can still be seen) from the granary above.  In 1566, Mary, Queen of Scots famously visited the wounded tenant of the castle, Lord Bothwell,"`
    - Token: `" Mary"`
    - Score: 15.312

16. Highest-activating token #16:

    - Excerpt from prompt: `" Richard III, Henry VIII and Charles I. It is most famous as the home of Catherine Parr (Henry VIII's widow) and her second husband, Thomas Seymour. Princess"`

- Token: `" Catherine"`
- Score: 15.275

17. Highest-activating token #17:

    - Excerpt from prompt: `" Peninsula\n\n#### Bodmin Moor\n\n#### Isles of Scilly\n\n#### St Mary's\n\n#### Tresco\n\n#### Bryher\n\n#### St Martin"`
    - Token: `" Mary"`
    - Score: 15.246

18. Highest-activating token #18:

    - Excerpt from prompt: `"'.\n\nOutside the cathedral's eastern end is the grave of the WWI heroine Edith"`
    - Token: `"ith"`
    - Score: 15.182

19. Highest-activating token #19:

    - Excerpt from prompt: `" many people visit for the region's literary connections; William Wordsworth, Beatrix Potter, Arthur Ransome and John Ruskin all found inspiration here.\n\n"`
    - Token: `"rix"`
    - Score: 15.135

20. Highest-activating token #20:

    - Excerpt from prompt: `" Peninsula\n\n#### Bodmin Moor\n\n#### Isles of Scilly\n\n#### St Mary's\n\n#### Tresco\n\n#### Bryher\n\n#### St Martin"`
    - Token: `" Mary"`
    - Score: 15.111

21. Highest-activating token #21:

    - Excerpt from prompt: `" _Mayor of Casterbridge_ locations hidden among modern Dorchester.  They include **Lucetta's House**, a grand Georgian affair with ornate door posts in Trinity St,"`
    - Token: `"etta"`
    - Score: 15.053

22. Highest-activating token #22:

    - Excerpt from prompt: `" leads down to this little cove and the remains of the small Tudor fort of **St Catherine's Castle**.\n\nPolkerris BeachBEACH\n\n( G"`
    - Token: `" Catherine"`
    - Score: 15.051

23. Highest-activating token #23:

    - Excerpt from prompt: `"-century **St Catherine's Lighthouse** and its 14th-century counterpart, **St Catherine's Or"`
    - Token: `" Catherine"`
    - Score: 14.979

24. Highest-activating token #24:

    - Excerpt from prompt: `" ) ; Castle Yard) stands behind a 15th-century gate near the church of St Mary de Castro ( MAP GOOGLE MAP ) ; Castle St),"`
    - Token: `" Mary"`
    - Score: 14.968

25. Highest-activating token #25:

   - Excerpt from prompt: `" the Glasgow School of Art.  It was there that he met the also influential artist and designer Margaret Macdonald, whom he married; they collaborated on many projects and were major influences on"`
   - Token: `" Margaret"`
   - Score: 14.930

26. Highest-activating token #26:

   - Excerpt from prompt: `" Nov-Mar )\n\nThe raising of the 16th-century warship the ⎵Mary Rose⎵ in 1982 was an extraordinary feat of marine archaeology.  Now the new £"`
   - Token: `"Mary"`
   - Score: 14.699

27. Highest-activating token #27:

   - Excerpt from prompt: `" was claimed by the Boleyn family and passed through the generations to Thomas, father of Anne Boleyn.  Anne was executed by her husband Henry VIII in 1533, who"`
   - Token: `" Anne"`
   - Score: 14.686

28. Highest-activating token #28:

   - Excerpt from prompt: `".  The village has literary cachet too { Wordsworth went to school here, and Beatrix Potter's husband, William Heelis, worked here as a solicitor for"`
   - Token: `"rix"`
   - Score: 14.658

29. Highest-activating token #29:

   - Excerpt from prompt: `" are William MacTaggart's Impressionistic Scottish landscapes and a gem by Thomas Millie Dow. There's also a special collection of James McNeill Whistler's lim"`
   - Token: `"ie"`
   - Score: 14.626

30. Highest-activating token #30:

   - Excerpt from prompt: `" Stay\n\nAMillgate House\n\nADevonshire Fell\n\nAHelaina\n\nAQuebecs\n\nALa Rosa Hotel\n\n## Yorkshire Highlights"`
   - Token: `"aina"`
   - Score: 14.578

The thirty lowest-activating tokens, along with the prompts from which they came, and their scores, are given below:

1. Lowest-activating token #1:

   - Excerpt from prompt: `" recounted the sighting of a disturbance in the loch by Mrs Aldie Mackay and her husband:  'There the creature disported itself, rolling and plunging for fully a minute"`
   - Token: `" husband"`
   - Score: -12.129

2. Lowest-activating token #2:

- Excerpt from prompt: `" paid time off work during menstruation\n• (often from male workers, who viewed the employment of women as competition) women should not be employed in"`
- Token: `" male"`
- Score: -11.344

3. Lowest-activating token #3:

  - Excerpt from prompt: `" family was devastated, but things quickly got worse.  Emily fell ill with tuberculosis soon after her brother's funeral; she never left the house again, and died on 19 December.  Anne"`
  - Token: `" brother"`
  - Score: -11.146

4. Lowest-activating token #4:

  - Excerpt from prompt: `" handsome Jacobean town house belonging to Shakespeare's daughter Susanna and her husband, respected doctor John Hall, stands south of the centre. The exhibition offers fascinating insights"`
  - Token: `" husband"`
  - Score: -11.016

5. Lowest-activating token #5:

  - Excerpt from prompt: `" hall was home to the 16th-century's second-most powerful woman, Elizabeth, Countess of Shrewsbury { known to all as Bess of Hardwick {"`
  - Token: `" Count"`
  - Score: -10.793

6. Lowest-activating token #6:

  - Excerpt from prompt: `" haunted places, with spectres from a phantom funeral to Lady Mary Berkeley seeking her errant husband.  Owner Sir Humphrey Wakefield has passionately restored the castle's extravagant medieval stater"`
  - Token: `" husband"`
  - Score: -10.682

7. Lowest-activating token #7:

  - Excerpt from prompt: `" Windsor Castle in 1861, Queen Victoria ordered its elaborate redecoration as a tribute to her husband.  A major feature of the restoration is the magnificent vaulted roof, whose gold mosaic"`
  - Token: `" husband"`
  - Score: -10.577

8. Lowest-activating token #8:

  - Excerpt from prompt: `"Ornate Plas Newydd was home to Lady Eleanor Butler and Miss Sarah Ponsonby, two society ladies who ran away from Ireland to Wales disguised as men, and"`
  - Token: `"onson"`
  - Score: -10.503

9. Lowest-activating token #9:

  - Excerpt from prompt: `" with DVD players, with tremendous views across the bay from the largest two.  Bridget and Derek really give this place a 'home away from home' ambience, and can arrange"`

- Token: " `Derek`"
- Score: -10.483

10. Lowest-activating token #10:
    - Excerpt from prompt: " `of adultery, debauchery, crime and edgy romance, and is filled with Chaucer's witty observations about human nature.\n\nHistory\n\nCanterbury's past`"
    - Token: "`cer`"
    - Score: -10.296

11. Lowest-activating token #11:
    - Excerpt from prompt: " `the city in 1645.  Legend has it that the disease-ridden inhabitants of **Mary King's Close** (a lane on the northern side of the Royal Mile on the site`"
    - Token: " `King`"
    - Score: -10.294

12. Lowest-activating token #12:
    - Excerpt from prompt: " `manor was founded in 1552 by the formidable Bess of Hardwick and her second husband, William Cavendish, who earned grace and favour by helping Henry VIII dissolve the English`"
    - Token: " `husband`"
    - Score: -10.251

13. Lowest-activating token #13:
    - Excerpt from prompt: " `Apartments** is the bedchamber where Mary, Queen of Scots gave birth to her son James VI, who was to unite the crowns of Scotland and England in 1603`"
    - Token: " `son`"
    - Score: -10.148

14. Lowest-activating token #14:
    - Excerpt from prompt: "`s at the behest of Queen Victoria, the monarch grieved here for many years after her husband's death.  Extravagant rooms include the opulent Royal Apartments and Dur`"
    - Token: " `husband`"
    - Score: -10.112

15. Lowest-activating token #15:
    - Excerpt from prompt: "`am-5pm Mar-Oct)\n\nThis ambitious three-dimensional interpretation of Chaucer's classic tales using jerky animatronics and audioguides is certainly entertaining`"
    - Token: "`cer`"
    - Score: -10.053

16. Lowest-activating token #16:
    - Excerpt from prompt: " `his death, in the hard-to-decipher Middle English of the day, Chaucer's _Tales_ is an unfinished series of 24 vivid stories told by a party`"
    - Token: "`cer`"
    - Score: -10.050

17. Lowest-activating token #17:
    - Excerpt from prompt: " `especially in **Poets' Corner**, where you'll find the resting places of Chaucer, Dickens, Hardy, Tennyson, Dr Johnson and Kipling, as well as`"

- Token: `"cer"`
- Score: -10.033

18. Lowest-activating token #18:
    - Excerpt from prompt: `" her?  She's up here saying his intent was this.\n\n¶ 35 Trujillo objected on the basis"`
    - Token: `" his"`
    - Score: -10.031

19. Lowest-activating token #19:
    - Excerpt from prompt: `" lived here happily with his sister Dorothy, wife Mary and three children John, Dora and Thomas until 1808, when the family moved to another nearby house at Allen Bank, and"`
    - Token: `" Thomas"`
    - Score: -9.934

20. Lowest-activating token #20:
    - Excerpt from prompt: `" home of Queen Isabella, who (allegedly) arranged the gruesome murder of her husband, Edward II.\n\nHoughton Hall"`
    - Token: `" husband"`
    - Score: -9.932

21. Lowest-activating token #21:
    - Excerpt from prompt: `" Saturday, four on Sunday).\n\nQueen Victoria bought Sandringham in 1862 for her son, the Prince of Wales (later Edward VII), and the features and furnishings remain"`
    - Token: `" son"`
    - Score: -9.883

22. Lowest-activating token #22:
    - Excerpt from prompt: `" the palace, which contains Mary's Bed Chamber, connected by a secret stairway to her husband's bedroom, and ends with the ruins of Holyrood Abbey.\n\nHoly"`
    - Token: `" husband"`
    - Score: -9.824

23. Lowest-activating token #23:
    - Excerpt from prompt: `" holidays.\n\nThe two-hour tour includes the **Throne Room**, with his-and-hers pink chairs initialed 'ER' and 'P'. Access is"`
    - Token: `" his"`
    - Score: -9.717

24. Lowest-activating token #24:
    - Excerpt from prompt: `" is packed with all manner of Highland memorabilia.  Look out for the secret portrait of Bonnie Prince Charlie { after the Jacobite rebellions all things Highland were banned, including pictures of"`
    - Token: `" Prince"`
    - Score: -9.691

25. Lowest-activating token #25:
    - Excerpt from prompt: `" the last college to let women study there; when they were finally admitted in 1988, some male students wore black armbands and flew the college flag at half mast.\n\n"`

- Token: `" male"`
- Score: -9.652

26. Lowest-activating token #26:

   - Excerpt from prompt: `"oh, Michael Bond's Paddington Bear, Beatrix Potter's Peter Rabbit, Roald Dahl's Willy Wonka and JK Rowling's Harry Potter are perennially popular"`
   - Token: `"ald"`
   - Score: -9.613

27. Lowest-activating token #27:

   - Excerpt from prompt: `" one of the rooms.  In 2003 the close was opened to the public as the Real Mary King's Close.\n\n### SCOTTISH PARLIAMENT BUILDING\n"`
   - Token: `" King"`
   - Score: -9.405

28. Lowest-activating token #28:

   - Excerpt from prompt: `", Mary, Dorothy and all three children. Samuel Taylor Coleridge's son Hartley is also buried here.\n\nGrasm"`
   - Token: `" Samuel"`
   - Score: -9.372

29. Lowest-activating token #29:

   - Excerpt from prompt: `", the town became northern Europe's most important pilgrimage destination, which in turn prompted Geoffrey Chaucer's ⎵The Canterbury Tales,⎵ one of the most outstanding works in English literature."`
   - Token: `"cer"`
   - Score: -9.351

30. Lowest-activating token #30:

   - Excerpt from prompt: `" Queen Isabella, who (allegedly) arranged the gruesome murder of her husband, Edward II.\n\nHoughton Hall"`
   - Token: `" Edward"`
   - Score: -9.272

## K.2  $n_{\text{SUBJ}}$ FEATURE VECTOR

The thirty highest-activating tokens, along with the prompts from which they came, and their scores, are given below:

1. Highest-activating token #1:

   - Excerpt from prompt: `"son Tower** and in front of it a beautiful statue of St Edmund by Dame Elisabeth Frink (1976).  The rest of the abbey spreads eastward like a r"`
   - Token: `"abeth"`
   - Score: 18.372

2. Highest-activating token #2:

   - Excerpt from prompt: `" a gorgeous hammerbeam roof and a striking sculpture of the crucified Christ by Dame Elisabeth Frink in the north transept.\n\nThe impressive entrance porch has a"`
   - Token: `"abeth"`

- Score: 17.388

3. Highest-activating token #3:
   - Excerpt from prompt: `" the elaborate Portuguese silver service or the impressive Egyptian service, a divorce present from Napoleon to Josephine"`
   - Token: `"ine"`
   - Score: 16.815

4. Highest-activating token #4:
   - Excerpt from prompt: `" rocky beach of **Priest's Cove**, while nearby are the ruins of **St Helen's Oratory**, supposedly one of the first Christian chapels built in West Cornwall"`
   - Token: `" Helen"`
   - Score: 16.309

5. Highest-activating token #5:
   - Excerpt from prompt: `", and opened in 1892, this brainchild of his Parisian actress wife, Josephine, was built by French architect Jules Pellechet to display a collection the Bow"`
   - Token: `"ine"`
   - Score: 16.267

6. Highest-activating token #6:
   - Excerpt from prompt: `" the film ˍBridget Jones's Diary;ˍ a local house was used as Bridget's parents' home.\n\n1Sights\n\nBroadway TowerTOWER"`
   - Token: `"idget"`
   - Score: 16.171

7. Highest-activating token #7:
   - Excerpt from prompt: `") by his side and a loyal band of followers in support.  Arthur went on to slay Rita Gawr, a giant who butchered"`
   - Token: `" Rita"`
   - Score: 16.079

8. Highest-activating token #8:
   - Excerpt from prompt: `" for the fact that Sir Robert Walpole's grandson sold the estate's splendid art collection to Catherine the Great of Russia to stave off debts { those paintings formed the foundation of the"`
   - Token: `" Catherine"`
   - Score: 16.039

9. Highest-activating token #9:
   - Excerpt from prompt: `" Highlights include the magnificent gold coach of 1762 and the 1910 Glass Coach (Prince William and Catherine Middleton actually used the 1902 State Landau for their wedding in 2011).\n\n"`
   - Token: `" Catherine"`
   - Score: 15.967

10. Highest-activating token #10:
    - Excerpt from prompt: `" by Canaletto, El Greco and Goya as well as 55 paintings by Josephine herself.  Among the 15,000 other objets d'art are incredible dresses from"`

- Token: `"ine"`
- Score: 15.906

11. Highest-activating token #11:

  - Excerpt from prompt: `" looks like something from a children's storybook (a fact not unnoticed by the author Antonia Barber, who set her much-loved fairy-tale _The Mousehole Cat"`
  - Token: `"ia"`
  - Score: 15.582

12. Highest-activating token #12:

  - Excerpt from prompt: `".  Precious little now remains save for a few nave walls, the ruined **St Mary's chapel**, and the crossing arches, which may"`
  - Token: `" Mary"`
  - Score: 15.443

13. Highest-activating token #13:

  - Excerpt from prompt: `".\n\nTrain\n\nThe northern terminus of the Welsh Highland Railway is on St Helen's Rd.  Trains run to Porthmadog (£35 return, 2½"`
  - Token: `" Helen"`
  - Score: 15.374

14. Highest-activating token #14:

  - Excerpt from prompt: `"2\n\n### KING RICHARD III\n\nIt's an amazing story.  Philippa Langley, a member of the Richard III Society, spent four-and-a"`
  - Token: `"a"`
  - Score: 15.358

15. Highest-activating token #15:

  - Excerpt from prompt: `" pit (which can still be seen) from the granary above.  In 1566, Mary, Queen of Scots famously visited the wounded tenant of the castle, Lord Bothwell,"`
  - Token: `" Mary"`
  - Score: 15.312

16. Highest-activating token #16:

  - Excerpt from prompt: `" Richard III, Henry VIII and Charles I. It is most famous as the home of Catherine Parr (Henry VIII's widow) and her second husband, Thomas Seymour. Princess"`
  - Token: `" Catherine"`
  - Score: 15.275

17. Highest-activating token #17:

  - Excerpt from prompt: `" Peninsula\n\n#### Bodmin Moor\n\n#### Isles of Scilly\n\n#### St Mary's\n\n#### Tresco\n\n#### Bryher\n\n#### St Martin"`
  - Token: `" Mary"`
  - Score: 15.246

18. Highest-activating token #18:

  - Excerpt from prompt: `"'.\n\nOutside the cathedral's eastern end is the grave of the WWI heroine Edith"`

- Token: `"ith"`
- Score: 15.182

19. Highest-activating token #19:

    - Excerpt from prompt: `" many people visit for the region's literary connections; William Wordsworth, Beatrix Potter, Arthur Ransome and John Ruskin all found inspiration here.\n\n"`
    - Token: `"rix"`
    - Score: 15.135

20. Highest-activating token #20:

    - Excerpt from prompt: `" Peninsula\n\n#### Bodmin Moor\n\n#### Isles of Scilly\n\n#### St Mary's\n\n#### Tresco\n\n#### Bryher\n\n#### St Martin"`
    - Token: `" Mary"`
    - Score: 15.111

21. Highest-activating token #21:

    - Excerpt from prompt: `" ⎵Mayor of Casterbridge⎵ locations hidden among modern Dorchester.  They include **Lucetta's House**, a grand Georgian affair with ornate door posts in Trinity St,"`
    - Token: `"etta"`
    - Score: 15.053

22. Highest-activating token #22:

    - Excerpt from prompt: `" leads down to this little cove and the remains of the small Tudor fort of **St Catherine's Castle**.\n\nPolkerris BeachBEACH\n\n( G"`
    - Token: `" Catherine"`
    - Score: 15.051

23. Highest-activating token #23:

    - Excerpt from prompt: `"-century **St Catherine's Lighthouse** and its 14th-century counterpart, **St Catherine's Or"`
    - Token: `" Catherine"`
    - Score: 14.979

24. Highest-activating token #24:

    - Excerpt from prompt: `" ) ; Castle Yard) stands behind a 15th-century gate near the church of St Mary de Castro ( MAP GOOGLE MAP ) ; Castle St),"`
    - Token: `" Mary"`
    - Score: 14.968

25. Highest-activating token #25:

    - Excerpt from prompt: `" the Glasgow School of Art.  It was there that he met the also influential artist and designer Margaret Macdonald, whom he married; they collaborated on many projects and were major influences on"`
    - Token: `" Margaret"`
    - Score: 14.930

26. Highest-activating token #26:

    - Excerpt from prompt: `" Nov-Mar )\n\nThe raising of the 16th-century warship the ⎵Mary Rose⎵ in 1982 was an extraordinary feat of marine archaeology.  Now the new £"`

- Token: `"Mary"`
- Score: 14.699

27. Highest-activating token #27:
    - Excerpt from prompt: `" was claimed by the Boleyn family and passed through the generations to Thomas, father of Anne Boleyn.  Anne was executed by her husband Henry VIII in 1533, who"`
    - Token: `" Anne"`
    - Score: 14.686

28. Highest-activating token #28:
    - Excerpt from prompt: `".  The village has literary cachet too { Wordsworth went to school here, and Beatrix Potter's husband, William Heelis, worked here as a solicitor for"`
    - Token: `"rix"`
    - Score: 14.658

29. Highest-activating token #29:
    - Excerpt from prompt: `" are William MacTaggart's Impressionistic Scottish landscapes and a gem by Thomas Millie Dow. There's also a special collection of James McNeill Whistler's lim"`
    - Token: `"ie"`
    - Score: 14.626

30. Highest-activating token #30:
    - Excerpt from prompt: `" Stay\n\nAMillgate House\n\nADevonshire Fell\n\nAHelaina\n\nAQuebecs\n\nALa Rosa Hotel\n\n## Yorkshire Highlights"`
    - Token: `"aina"`
    - Score: 14.578

The thirty lowest-activating tokens, along with the prompts from which they came, and their scores, are given below:

1. Lowest-activating token #1:
   - Excerpt from prompt: `" family was devastated, but things quickly got worse.  Emily fell ill with tuberculosis soon after her brother's funeral; she never left the house again, and died on 19 December.  Anne"`
   - Token: `" brother"`
   - Score: -11.732

2. Lowest-activating token #2:
   - Excerpt from prompt: `" recounted the sighting of a disturbance in the loch by Mrs Aldie Mackay and her husband:  'There the creature disported itself, rolling and plunging for fully a minute"`
   - Token: `" husband"`
   - Score: -11.608

3. Lowest-activating token #3:
   - Excerpt from prompt: `" paid time off work during menstruation\n• (often from male workers, who viewed the employment of women as competition) women should not be employed in"`
   - Token: `" male"`

- Score: -11.324

4. Lowest-activating token #4:

   - Excerpt from prompt: `"Ornate Plas Newydd was home to Lady Eleanor Butler and Miss Sarah Ponsonby, two society ladies who ran away from Ireland to Wales disguised as men, and"`
   - Token: `"onson"`
   - Score: -11.228

5. Lowest-activating token #5:

   - Excerpt from prompt: `" of adultery, debauchery, crime and edgy romance, and is filled with Chaucer's witty observations about human nature.\n\nHistory\n\nCanterbury's past"`
   - Token: `"cer"`
   - Score: -11.007

6. Lowest-activating token #6:

   - Excerpt from prompt: `" Apartments** is the bedchamber where Mary, Queen of Scots gave birth to her son James VI, who was to unite the crowns of Scotland and England in 1603"`
   - Token: `" son"`
   - Score: -10.971

7. Lowest-activating token #7:

   - Excerpt from prompt: `" handsome Jacobean town house belonging to Shakespeare's daughter Susanna and her husband, respected doctor John Hall, stands south of the centre. The exhibition offers fascinating insights"`
   - Token: `" husband"`
   - Score: -10.884

8. Lowest-activating token #8:

   - Excerpt from prompt: `" his death, in the hard-to-decipher Middle English of the day, Chaucer's _Tales_ is an unfinished series of 24 vivid stories told by a party"`
   - Token: `"cer"`
   - Score: -10.854

9. Lowest-activating token #9:

   - Excerpt from prompt: `" especially in **Poets' Corner**, where you'll find the resting places of Chaucer, Dickens, Hardy, Tennyson, Dr Johnson and Kipling, as well as"`
   - Token: `"cer"`
   - Score: -10.794

10. Lowest-activating token #10:

    - Excerpt from prompt: `"am-5pm Mar-Oct)\n\nThis ambitious three-dimensional interpretation of Chaucer's classic tales using jerky animatronics and audioguides is certainly entertaining"`
    - Token: `"cer"`
    - Score: -10.793

11. Lowest-activating token #11:

    - Excerpt from prompt: `" haunted places, with spectres from a phantom funeral to Lady Mary Berkeley seeking her errant husband. Owner Sir Humphrey Wakefield has passionately restored the castle's extravagant medieval stater"`

- Token: " husband"
- Score: -10.696

12. Lowest-activating token #12:

    - Excerpt from prompt: " Windsor Castle in 1861, Queen Victoria ordered its elaborate redecoration as a tribute to her husband. A major feature of the restoration is the magnificent vaulted roof, whose gold mosaic"
    - Token: " husband"
    - Score: -10.673

13. Lowest-activating token #13:

    - Excerpt from prompt: " hall was home to the 16th-century's second-most powerful woman, Elizabeth, Countess of Shrewsbury { known to all as Bess of Hardwick {"
    - Token: " Count"
    - Score: -10.617

14. Lowest-activating token #14:

    - Excerpt from prompt: " Saturday, four on Sunday).\n\nQueen Victoria bought Sandringham in 1862 for her son, the Prince of Wales (later Edward VII), and the features and furnishings remain"
    - Token: " son"
    - Score: -10.556

15. Lowest-activating token #15:

    - Excerpt from prompt: " is packed with all manner of Highland memorabilia. Look out for the secret portrait of Bonnie Prince Charlie { after the Jacobite rebellions all things Highland were banned, including pictures of"
    - Token: " Prince"
    - Score: -10.424

16. Lowest-activating token #16:

    - Excerpt from prompt: " beautiful, time-worn rooms hold fascinating relics, including the cradle used by Mary for her son, James VI of Scotland (who also became James I of England), and fascinating letters"
    - Token: " son"
    - Score: -10.266

17. Lowest-activating token #17:

    - Excerpt from prompt: ", the town became northern Europe's most important pilgrimage destination, which in turn prompted Geoffrey Chaucer's ˍThe Canterbury Tales,ˍ one of the most outstanding works in English literature."
    - Token: "cer"
    - Score: -10.250

18. Lowest-activating token #18:

    - Excerpt from prompt: " the city in 1645. Legend has it that the disease-ridden inhabitants of **Mary King's Close** (a lane on the northern side of the Royal Mile on the site"
    - Token: " King"
    - Score: -10.177

19. Lowest-activating token #19:

- Excerpt from prompt: `" with DVD players, with tremendous views across the bay from the largest two. Bridget and Derek really give this place a 'home away from home' ambience, and can arrange"`
- Token: `" Derek"`
- Score: -10.124

20. Lowest-activating token #20:

    - Excerpt from prompt: `" her? She's up here saying his intent was this.\n\n¶ 35 Trujillo objected on the basis"`
    - Token: `" his"`
    - Score: -10.113

21. Lowest-activating token #21:

    - Excerpt from prompt: `" the last college to let women study there; when they were finally admitted in 1988, some male students wore black armbands and flew the college flag at half mast.\n\n"`
    - Token: `" male"`
    - Score: -10.058

22. Lowest-activating token #22:

    - Excerpt from prompt: `"s at the behest of Queen Victoria, the monarch grieved here for many years after her husband's death. Extravagant rooms include the opulent Royal Apartments and Dur"`
    - Token: `" husband"`
    - Score: -10.018

23. Lowest-activating token #23:

    - Excerpt from prompt: `" home of Queen Isabella, who (allegedly) arranged the gruesome murder of her husband, Edward II.\n\nHoughton Hall"`
    - Token: `" husband"`
    - Score: -9.989

24. Lowest-activating token #24:

    - Excerpt from prompt: `", Van Dyck, Vermeer, El Greco, Poussin, Rembrandt, Gainsborough, Turner, Constable, Monet, Pissarro,"`
    - Token: `"brand"`
    - Score: -9.937

25. Lowest-activating token #25:

    - Excerpt from prompt: `" 24 vivid stories told by a party of pilgrims journeying between London and Canterbury. Chaucer successfully created the illusion that the pilgrims, not Chaucer (though he appears in the"`
    - Token: `"cer"`
    - Score: -9.909

26. Lowest-activating token #26:

    - Excerpt from prompt: `" the palace, which contains Mary's Bed Chamber, connected by a secret stairway to her husband's bedroom, and ends with the ruins of Holyrood Abbey.\n\nHoly"`
    - Token: `" husband"`
    - Score: -9.862

27. Lowest-activating token #27:
    - Excerpt from prompt: `" lived here happily with his sister Dorothy, wife Mary and three children John, Dora and Thomas until 1808, when the family moved to another nearby house at Allen Bank, and"`
    - Token: `" Thomas"`
    - Score: -9.842

28. Lowest-activating token #28:
    - Excerpt from prompt: `" 19 prime ministers, countless princes, kings and maharajahs, famous explorers, authors and"`
    - Token: `" prime"`
    - Score: -9.733

29. Lowest-activating token #29:
    - Excerpt from prompt: `" held court in the Palace of Holyroodhouse for six brief years, but when her son James VI succeeded to the English throne in 1603, he moved his court to London"`
    - Token: `" son"`
    - Score: -9.711

30. Lowest-activating token #30:
    - Excerpt from prompt: `", Mary, Dorothy and all three children. Samuel Taylor Coleridge's son Hartley is also buried here.\n\nGrasm"`
    - Token: `" Samuel"`
    - Score: -9.654

