# OpenReview forum: "Observable Propagation: Uncovering Feature Vectors in Transformers"
_ICLR.cc/2024/Conference — ICLR 2024 Conference Withdrawn Submission_

### Official Review · Reviewer_VvVc · 2023-10-31

**Soundness:** 2 fair
**Presentation:** 1 poor
**Contribution:** 2 fair
**Rating:** 3
**Confidence:** 2

**Summary:**

The paper proposes a method to compute a "feature vector" in transformers for a given task.

**Strengths:**

The method requires no labeled training data, in contrast to other similarly motivated methods like probing.

**Weaknesses:**

I find the paper quite opaque. The paper should directly give a formal definition of an "observable", state how it is used/useful, and show how it is computed. Instead, the paper gives a long narrative of several motivations in words. The theorem about how the layer norm doesn't substantially the direction of feature vectors seems pretty specific to the narrative here; it is hard to see much general value out of the context.

I'm not very knowledgeable in the area of gender bias analysis so I'm not qualified to make comments on the value of the experiments. But given my lack of confidence on the value of the technical part, it is hard for me to see the value of the experiments either.

**Questions:**

N/A

---

> ### Author Response · Authors · 2023-11-20
>
> Thank you very much for your review, and for recognizing the value of observable propagation in allowing for feature vectors to be found. With regard to your comments, we address them individually below:
> * “The paper should directly give a formal definition of an "observable", state how it is used/useful, and show how it is computed. Instead, the paper gives a long narrative of several motivations in words.”
>     * We have added a formal definition and explanation of an observable in Appendix A. To summarize, an observable is a linear functional from $\mathbb{R}^{\mathtt{d_vocab}} \to \mathbb{R}$, where $\mathtt{d_vocab}$ is the size of the model vocabulary. We refer to the action of taking the inner product between the model’s logits outputs and an observable n as getting the output of the model with respect to observable n. Intuitively, observables correspond to tasks on which we want to measure model output. As an example, for the subject pronoun prediction task, we want to measure the extent to which the model’s logit prediction for the pronoun “she” is greater than the logit for the pronoun “he”. As such, an observable corresponding to this task is given by $e_{\mathtt{” she”}} - e_{\mathtt{” he”}}$ where $e_{\mathtt{token}}$ is a one-hot vector of size $\mathtt{d_vocab}$ with a one in the position corresponding to \texttt{token}. This is because the inner product between this observable and the model’s predicted logits vector precisely yields the logit for the pronoun “she” minus the logit for the pronoun “he”. We hope that this explanation has better clarified our approach.
> * “The theorem about how the layer norm doesn't substantially the direction of feature vectors seems pretty specific to the narrative here; it is hard to see much general value out of the context.”
>     * The reason why this theorem is important is because it implies that the general approach of finding feature vectors — something considered critical to the mechanistic interpretability community (e.g. please see Olah (2022), “finding and understanding interpretable neurons [a type of feature vector] […] isn’t just one of many interesting questions. Arguably, it’s the central task [of mechanistic interpretability]”) — is a useful means of understanding how Transformers process information, even in the presence of ubiquitous LayerNorm nonlinearities. After all, if LayerNorms substantially affected the behavior of feature vectors, then this would imply that the entire idea of using linear feature vectors is ill-founded, because nonlinear LayerNorm effects would dominate the behavior of the model, even in linear attention-only circuits. Thus, being able to show that this is not the case provides theoretical backing for this entire paradigm.
>
> We hope that we have been able to successfully address your comments; if you have any further questions, please do not hesitate to let us know. Again, thank you for your review.
>
> Reference: Chris Olah (2022) Mechanistic Interpretability, Variables, and the Importance of Interpretable Bases. https://transformer-circuits.pub/2022/mech-interp-essay/index.html

---

### Official Review · Reviewer_Gf3o · 2023-11-01

**Soundness:** 1 poor
**Presentation:** 4 excellent
**Contribution:** 2 fair
**Rating:** 3
**Confidence:** 3

**Summary:**

This paper proposes a method called Observable Propagation (ObProp) to identify feature vectors in large language models. Because of the success of LLMs, interpreting what their intermediate layers learn has turned out to be an emerging topic. The authors propose that linear transforms of language model logits, which they call observables, can be captured approximately via feature vectors that can be computed by a simple forward model through the model.

In more detail, the basic idea is to define an abstract vector called observable n and feature vector y, such that the inequality n.f(x) = y.x is satisfied for a non-linearity f (an intermediate layer of the LLM). This inequality cannot be exact but the authors propose to compute an approximation by approximating each neural non-linearity by its gradient (Taylor approximation) and each attention mechanism by the attention weights (which are assumed to be constants). This lets them learn a feature vector y for each observable n.

The authors propose that studying the norms of such vectors and looking at their correlation (a coupling coefficient) will suggest us more about the observables. Simple experiments using GPT-Neo-1.3B suggest that such learnt feature vectors for the gendered pronounds prediction weakly match the ones recovered by looking at logit differences (via path patching). Additional experiments on occupational gender bias predicts that LLMs do exhibit gender-occupation bias.

**Strengths:**

- Mechanistic interpretability is an important research direction and the authors propose a simple method towards this research direction which does not rely on large-scale compute or data.

- Their experiments, including App. H, suggest that the ObProp algorithm does seem to recover meaningful interpretation of various attention heads that occur in the model, which could then be analyzed more rigorously using more powerful methods.

**Weaknesses:**

- This method seems like an oversimplification of the architecture of an LLM. The ObProp algorithm propagates through the layers by approximating almost all non-linearities by their first-order gradients.

- Moreover, the work abstracts out QK circuits in LLMs, i.e. the attention scores are treated as constants and not as non-linearities, therefore the problem is oversimplified and I'm not sure the theorems shown are necessarily that insightful. As the authors note, their method ObProp does not capture the action of induction heads.

- Theorem 1 proposes that layer norms do not affect the feature vectors, by showing that the expected cosine similarity is close to 0. However, in this theorem, the activations and observables x, n are chosen to Gaussian with mean 0 and covariance I. What's the point of this assumption? And how do the authors conclude that the theorem 1, as stated with these distributions, implies their claim about layer norm for general activations?

- Related to the above, why did the authors look at uniform distributions on the sphere in theorem 2? Even if the vectors have bounded norm, why do the authors assume uniformity?

**Questions:**

Some questions were raised above.

- Typo in the equation in 2.2 and below, score has both 1 and 2 subscripts.

---

> ### Author Response · Authors · 2023-11-21
> **Response to review part 1/2**
>
> Thank you for your detailed review and comments.
> We are happy to see that you recognize the ability of observable propagation to recover meaningful feature vectors, and the broader implications that this has for the field of mechanistic interpretability.
> We will address your comments individually below.
>
> > “This method seems like an oversimplification of the architecture of an LLM. The ObProp algorithm propagates through the layers by approximating almost all non-linearities by their first-order gradients.”
>
> We find that oftentimes, useful circuits in Transformers involve either very few nonlinearities or none; additionally, these nonlinearities often display local linearity in input regions that we care about. For example, the circuit corresponding to the `attn6::6` (head 6, layer 6) feature vector for the subject pronoun case contains only attention sublayers; this is also the case for the circuits corresponding to the feature vectors used in the recently-updated political party prediction experiment and the C vs. Python classification experiment.
>
> As for circuits that contain MLP sublayers: we investigated the accuracy of the first-order linear approximation of the circuit consisting of `MLP16 → MLP18 → final LayerNorm → unembeddings` on the occupation prediction dataset; this is because this subcircuit is the nonlinear component of the circuit for the `attn6::6` feature vector for the occupational bias observable. We found that on this dataset, the root mean squared error of the linear approximation was approximately 0.4139 logits, which is only approximately 3.5% of the mean output value of this circuit. This suggests that, at least on specific tasks with constrained input distributions, first-order approximations of MLP nonlinearities suffice. We have updated Appendix D with further information.
>
> We recognize that the observable propagation algorithm does indeed involve a simplification, particularly in its approximation of MLP nonlinearities using first-order gradients. This approach was chosen to strike a balance between computational tractability and analytical insight. However, the method's utility *extends beyond these initial simplifications*. The paradigm introduced by observable propagation is a preliminary step towards more comprehensive mechanistic analyses of MLP nonlinearities. By enabling the backward propagation of feature vectors through the network, our method provides a novel framework to explore how MLP nonlinearities interact with different **tasks**, not just with varying inputs.
>
> We believe that the position of observable propagation in its current state can be compared to the seminal transformers circuits work done by [Elhage et al. 2021](https://transformer-circuits.pub/2021/framework/index.html), which ignored MLP layers entirely. The assumptions made in that work were far more simplifying than our treatment of MLP linearization, but time has demonstrated that their work has born substantial fruit in pushing forward our knowledge of mechanistic interpretability of complete models. Indeed, the recent work by [Bricken et al. 2023](https://transformer-circuits.pub/2023/monosemantic-features/index.html) on dictionary learning of features takes a similar approach by only analyzing one-layer transformers, a far more simplified scenario than ours, with this same aim of later providing more complete insights. We hope that our work can do the same.
>
> > “Moreover, the work abstracts out QK circuits in LLMs, i.e. the attention scores are treated as constants and not as non-linearities, therefore the problem is oversimplified and I'm not sure the theorems shown are necessarily that insightful. As the authors note, their method ObProp does not capture the action of induction heads.”
>
> Even though we do not address QK circuits with this method, we still believe that our method’s application to OV circuits is highly valuable. If the QK circuit determines where information comes from in a Transformer, the OV circuit determines how that information is transformed throughout the model – and in many relevant tasks, these computations in the OV circuit are highly non-trivial. Using the occupational gender bias task that we address in our paper as an example: the fact that the model uses the same features to predict occupations as it does to predict gendered pronouns is certainly not obvious a priori; this is just one example of the insights that can be gained by looking at OV circuits. Furthermore, even if complex machinery in the model’s QK circuits (such as induction heads) is used to transfer information between tokens, nevertheless, whenever we want to understand what sort of information the model uses to predict one token as opposed to another, the answer to this question lies in the model’s OV circuits. And observable propagation can be used to reveal this answer.

---

> ### Author Response · Authors · 2023-11-21
> **response to review part 2/2**
>
> > “Theorem 1 proposes that layer norms do not affect the feature vectors, by showing that the expected cosine similarity is close to 0. However, in this theorem, the activations and observables x, n are chosen to Gaussian with mean 0 and covariance I. What's the point of this assumption? And how do the authors conclude that the theorem 1, as stated with these distributions, implies their claim about layer norm for general activations?
>
> We appreciate the reviewer's critical assessment of Theorem 1. Upon reevaluation, we acknowledge that the Gaussian assumption for activations may not be necessary for the theorem's proof. Consequently, we have revised the theorem's statement to remove this assumption, thus broadening its applicability.
>
> Regarding the Gaussian assumption for observables, this choice was guided by the nascent nature of the observable concept in this field. Given the lack of established intuition or precedent for the distribution of observables, we opted for a Gaussian model as a starting point, owing to its mathematical tractability and prevalence in theoretical studies. This assumption, however, is not intended to limit the theorem's scope. Empirically, we observed that Theorem 1's conclusions hold true across a range of scenarios: the cosine similarities of feature vectors obtained by directly linearly-approximating LayerNorms with feature vectors obtained by ignoring LayerNorms are all extremely high. These observations are detailed in Appendix I and provide practical evidence supporting the theorem's broader implications.
>
> In summary, while the initial version of Theorem 1 employed specific assumptions for mathematical convenience, our empirical findings and subsequent theorem revision reinforce its relevance and potential applicability to a broader range of activation and observable distributions.
>
> >Related to the above, why did the authors look at uniform distributions on the sphere in theorem 2? Even if the vectors have bounded norm, why do the authors assume uniformity?”
>
> Similar to our above response with regard to Theorem 1, note that we empirically observe that the coupling coefficients are accurate estimators of the constant of proportionality between the dot products of activations with two given feature vectors – please see Table 2 in Section 4.1. Thus, this theorem as well serves to theoretically motivate this quantity – that is, the coupling coefficient – which empirically bears out the behavior that is predicted by the theorem.
>
> We hope that we have been able to demonstrate the theoretical soundness of observable propagation with regard to your questions. If you have any further comments, please do not hesitate to let us know. Again, thank you for your review.

---

### Official Review · Reviewer_DiKb · 2023-11-05

**Soundness:** 2 fair
**Presentation:** 3 good
**Contribution:** 2 fair
**Rating:** 5
**Confidence:** 3

**Summary:**

The authors propose a method ObsProb  to better understand which parts (paths) of neural networks in neural networks contribute to which predictions. The idea is to start with a linear functional on the logits layer and then give some (approximate) mathematical arguments as to how this can be propagated down the network.

Empirically the authors do a case study with gender bias. They show that the n_subj (he/she) and n_obj (her/him) feature vectors derived from ObsProb have high cosine similarities for 4 heads and these cosine similarities are vary high. This indicates that according to the authors' approach, the same underlying features of the network are being used for these predictions. Moreover, the authors find a similar result with n_{bias} (which is an observable for occupational bias).  They find a particular path for n_{subj} has a very high correlation with that of n_{bias}.

**Strengths:**

Interpretability is complex and challenging for LLMs. Being able to understand how different predictions are made my similar parts of the model could be very useful. The authors' results are around gender and occupational bias using their approach are insightful.

**Weaknesses:**

The empirical evaluation is largely a case study and doesn't feel rigorous. It would be more convincing to have a more quantitative evaluation across more types of observables compared to a baseline, along with the case study.

Nit: It would be good to define the term "unembedding matrix".

**Questions:**

My main concerns with the approach are around rigorous evaluation as discussed above.

---

> ### Author Response · Authors · 2023-11-20
>
> Thank you for taking the time to review our work. We are glad to hear that you appreciate the power of observable propagation for obtaining an understanding of model internals, particularly in the gender bias case. With regard to your thoughts on evaluating our method: as per your suggestion, we ran substantial additional experiments involving different observables and performed quantitative comparisons to feature vectors obtained by linear/logistic regression. We evaluated ObProp’s performance across a broader variety of tasks, including subject pronoun prediction, identifying American politicians’ party affiliations, and distinguishing between C and Python code. In particular, feature vectors found via ObProp match or exceed performance of feature vectors found using methods that require training.
>
> We updated the paper with these results in Section 4.3. In general, we found that for the gendered pronoun prediction setting, the feature vector obtained by linear regression was only able to match the performance of the observable propagation feature vector when the linear regression was trained on 60 embeddings. Note that, in contrast, our observable propagation feature vectors could be obtained without running any forward passes. For the Python vs. C code setting, the logistic regression feature vector only matched the performance of the observable propagation feature vector when the logistic regression was trained on 50 embeddings. And for the political party identification setting, even when trained on 3/4ths of an artificial dataset of politicians (60 embeddings in the training set), the linear regression feature vector yielded far worse results than the observable propagation feature vector (correlation coefficient $r^2\approx0.30$ versus $r^2\approx0.43$). Our findings suggest that observable propagation offers a notable advantage over traditional methods, particularly in the low-data regime.
>
> If you have any further questions or comments, please do not hesitate to let us know. Again, thank you for your review.
>
> (P.S. In response to your comments, we also added a brief explanation that an unembedding matrix is the matrix “which projects the model's final activations into logits space”, in the place where the term is first used.)

---

> > ### Comment · Reviewer_DiKb · 2023-11-22
> >
> > I am sorry but I don't see an updated paper? (Not sure what the rebuttal rules are and whether that is allowed or not)

---

> > > ### Author Response · Authors · 2023-11-23
> > >
> > > The updated paper is available using the same PDF link next to the paper title; this link now redirects to the updated version.
> > >
> > > With regard to the rebuttal rules regarding paper revision, note that authors are allowed an unlimited number of revisions during the duration of the rebuttal period. From the Call for Papers: “Once the reviews are posted, authors are free to upload modifications to the paper during the discussion period.” Additionally, from the Author Guide:
> > >
> > > > Q. For rebuttal revisions, are we limited to one upload or can we update the paper several times?
> > >
> > > > You can upload revisions until Nov 22 (End of Day, Anywhere on Earth), but reviewers and area chairs are not required to look at every revision. It is up to you to clearly communicate what’s been changed.
> > >
> > > We hope that this clarifies things for you. As always, please let us know if you have any further questions.